# Forecasting extremely high ischemic stroke incidence using meteorological time serie

**Lucia Babalova**[1☯], **Marian Grendar**[2,3☯], **Egon Kurca**[1], **Stefan Sivak**[1], **Ema Kantorova**[1], **Katarina Mikulova**[4], **Pavel Stastny**[4], **Pavel Fasko**[4], **Kristina Szaboova**[4], **Peter Kubatka**[5], **Slavomir Nosal**[6], **Robert Mikulik**[7,8], **Vladimir Nosal**[1]*

1 Clinic of Neurology, Jessenius Faculty of Medicine in Martin, Comenius University in Bratislava, Bratislava, Slovakia, 2 Laboratory of Bioinformatics and Biostatistics, Biomedical Centre Martin, Jessenius Faculty of Medicine in Martin, Comenius University in Bratislava, Bratislava, Slovakia, 3 Laboratory of Theoretical Methods, Institute of Measurement Science, Slovak Academy of Sciences, Bratislava, Slovakia, 4 Slovak Hydrometeorological Institute in Bratislava, Bratislava, Slovakia, 5 Department of Medical Biology, Jessenius Faculty of Medicine in Martin, Comenius University in Bratislava, Bratislava, Slovakia, 6 Clinic of Paediatric Anaesthesiology and Intensive Medicine, Jessenius Faculty of Medicine in Martin, Comenius University in Bratislava, Bratislava, Slovakia, 7 First Department of Neurology, Faculty of Medicine, Masaryk University, Brno, Czech Republic, 8 Neurology Department, Tomas Bata Regional Hospital, Zlín, Czech Republic

☯ These authors contributed equally to this work.
* vladimir.nosal@uniba.sk

**Data Availability Statement:** All relevant data are within the paper and its Supporting information files.

## Abstract

### Motivation

The association between weather conditions and stroke incidence has been a subject of interest for several years, yet the findings from various studies remain inconsistent. Additionally, predictive modelling in this context has been infrequent. This study explores the relationship of extremely high ischaemic stroke incidence and meteorological factors within the Slovak population. Furthermore, it aims to construct forecasting models of extremely high number of strokes.

### Methods

Over a five-year period, a total of 52,036 cases of ischemic stroke were documented. Days exhibiting a notable surge in ischemic stroke occurrences (surpassing the 90th percentile of historical records) were identified as extreme cases. These cases were then scrutinized alongside daily meteorological parameters spanning from 2015 to 2019. To create forecasts for the occurrence of these extreme cases one day in advance, three distinct methods were employed: Logistic regression, Random Forest for Time Series, and Croston's method.

### Results

For each of the analyzed stroke centers, the cross-correlations between instances of extremely high stroke numbers and meteorological factors yielded negligible results. Predictive performance achieved by forecasts generated through multivariate logistic regression and Random Forest for time series analysis, which incorporated meteorological data, was on par with that of Croston's method. Notably, Croston's method relies solely on the stroke time series data. All three forecasting methods exhibited limited predictive accuracy.

**Funding:** The author(s) received no specific funding for this work.

**Competing interests:** The authors have declared that no competing interests exist.

**Abbreviations:** AIC, Akaike Information Criterion; AUC, Area under ROC; BA, Bratislava; BB, Banska Bystrica; BIC, Bayesian Information Criterion; CCF, Cross Correlation Function; DI, Discomfort Index; EDA, Exploratory Data Analysis; *Ext*, Time series of extremely high number of stroke counts; ETV, Effective Temperature Taking Wind Velocity; GLM, Generalized Linear Model; ICD, International Classification of Diseases; KE, Kosice; MT, Martin; RFTS, Random Forest for Time Series; ROC, Receiver Operation Characteristics Curve; VCI, Wind Chill Index.

## Conclusions

The task of predicting days characterized by an exceptionally high number of strokes proved to be challenging across all three explored methods. The inclusion of meteorological parameters did not yield substantive improvements in forecasting accuracy.

## Introduction

The exploration of the association between stroke incidence, mortality, and environmental factors has spanned several decades. Initial observations by [1] and a study by [2] in the early 1980s first unveiled a seasonal pattern in stroke mortality. Notably, clinical observations revealing clusters of daily stroke admissions, as highlighted by [3], provided further impetus to investigate the intricate connections between weather and stroke occurrences. Over time, this research has encompassed a global scale, yielding a plethora of results that, regrettably, remain inconsistent. While some researchers (e.g., [1–12]) have reported correlations between stroke incidents and environmental factors, others (e.g., [13, 14]) have failed to substantiate such links. Similar inconsistencies have emerged from studies concerning stroke seasonality (e.g., [5, 14–16]).

Numerous meta-analyses (e.g., [17–19]) have attempted to consolidate these findings in recent years. However, the divergence in study designs, utilized databases, statistical methodologies, and varying climate conditions across countries has hindered the direct comparison of results. As the meta-analysis of seasonal patterns by [19] revealed, climatic factors appear to exert a substantial influence on seasonality. Furthermore, the scope and duration of data collection prove crucial; while some studies (e.g., [2–4, 7, 9, 13–15]) focus on localized hospital or regional data, only a handful (e.g., [5, 16]) adopt a nationwide perspective.

Understanding the intricate relationship between environmental factors and stroke incidents bears significant potential for preventive interventions. Early investigations by [3] suggested heightened stroke risk on warm days, emphasizing the importance of maintaining adequate fluid intake and considering antiplatelet therapy as preventive measures. However, subsequent studies (e.g., [13, 15, 20]) contradicted these findings, challenging the notion of an increased stroke risk during warmer temperatures. In contrast, observations by [9, 16, 20–22] implicated winter and colder days in heightened stroke incidence. The conflicting outcomes have impeded the formulation of unequivocal preventive recommendations.

To address this conundrum, we propose a shift from mere modeling to forecasting as a novel approach. Historically, most studies have centered on modeling the relationship between weather and stroke counts, reporting statistically significant meteorological variables. However, only a handful of studies (e.g., [23], and partially [24]) have ventured into forecasting stroke counts based on meteorological variables. Importantly, the objectives of quantifying statistical associations through significance testing diverge from those of forecasting. As recognized in time series literature [25–27], statistical significance testing possesses limited utility in forecasting. In time series forecasting the attention is paid 'to finding models that forecasts well'; [26].

With this perspective, in our study we augment traditional association modeling by incorporating stroke count forecasting. Our primary contribution is twofold:

- Evaluation of Predictive Methods: We systematically evaluate the predictive efficacy of different forecasting methods, including Generalized Linear Regression (GLM), RandomForest for Time Series, and Croston's method, in predicting stroke counts.

- Comparison of Forecasting Approaches: We compare forecasts that leverage meteorological time series data against those based solely on historical stroke count data. This comparison aims to determine whether incorporating meteorological data improves the accuracy and utility of stroke count predictions.

In the pursuit of making our approach clinically relevant, we chose to focus on forecasting extremely high stroke counts. This facet holds practical significance, as it enables clinic administrators to adjust resources and allocate clinicians effectively to accommodate predicted surges in stroke cases. By accurately forecasting these extreme events, healthcare providers can be better prepared to manage increased patient loads, ensuring that adequate care is available during critical periods.

Notably, forecasting high stroke counts sidesteps the challenge of selecting a quality measure for predictions. By transforming raw stroke counts into binary time series (ordinary stroke count/high stroke count), we can employ sensitivity and specificity metrics to quantify forecast quality. Our comparative analysis involves three forecasting methods: multivariate logistic regression, Random Forest for Time Series analysis, and Croston's method. While the former two incorporate meteorological time series, the latter relies solely on historical stroke counts. Croston's method serves as a benchmark, facilitating an evaluation of the contribution of meteorological data to forecast improvements. Furthermore, alongside forecasting endeavors, we employ multivariate logistic regression and Random Forest for Time Series analysis to undertake conventional association modeling.

The dataset employed in this study originates from the primary stroke centers in the Slovak Republic, with only a limited number of studies addressing this topic in Central Europe (e.g., [7, 9, 28]). Notably, the majority of prior research originates from countries characterized by different climate conditions (e.g., [1–5, 8, 13–16, 20–22]).

In summary, the complex interplay between stroke incidence, meteorological factors, and subsequent contradictory findings necessitates novel approaches. By shifting the focus from association modeling to forecasting, we aim to provide fresh insights into the intricate relationships and practical implications for stroke prevention and resource allocation.

## Methods

### Patients

Slovakia, situated in Central Europe, boasts a population of slightly over 5.4 million as of 2021 [29]. The country is divided into 79 administrative districts. Daily stroke incidence data for the entire period spanning from January 2015 to December 2019 were extracted from the databases of all health insurance companies in Slovakia. The date when the data were accessed for research purposes was May 24, 2021.

Inclusion criteria consisted of patients diagnosed with ischaemic stroke, identified by the ICD-10 code I63. The diagnosis was predicated upon the updated stroke definition established by the American Heart Association/American Stroke Association, as outlined in [30]. To pinpoint patients' district affiliations, their places of residence were employed. Every patient underwent either a brain CT or MRI, aiding in the determination of the day of onset. This day was anchored to the brain imaging examination date and the date of ambulance transportation culminating in hospitalization.

In the pursuit of comprehensiveness, patients of all age groups with ischaemic stroke were encompassed within the study cohort. This study was conducted as part of a research project approved by the Ethics Committee of the Jessenius Faculty of Medicine at Comenius

Slovakia map of Köppen climate classification

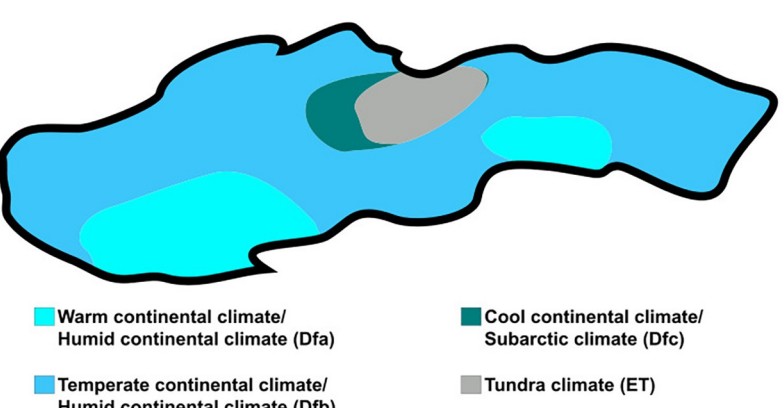

■ Warm continental climate/
Humid continental climate (Dfa)

■ Temperate continental climate/
Humid continental climate (Dfb)

■ Cool continental climate/
Subarctic climate (Dfc)

■ Tundra climate (ET)

**Fig 1. Köppen-Geiger's climate classification of Slovakia.** Source: Wikimedia Commons.

University in Bratislava. All methodologies adhered to pertinent guidelines and regulations. All patient data were fully anonymized.

## Meteorological data

Alisov's climate classification [31] categorizes Slovakia in the continental climate of the temperate belt with four seasons (winter, spring, summer, autumn). According to Köppen-Geiger's climate classification [32], the warm-summer humid continental climate prevails in Slovakia; see Fig 1.

The observations of the Slovak Hydrometeorological Institute were used. The database consists of the daily climatological parameters from a nationwide network of meteorological stations; see Climatological Station Network in Slovakia. Stations were assigned to districts according to geographical location, so the place of the residence of patients is paired with meteorological station.

The dataset encompasses daily parameters, including minimum temperature ($t_{min}$), maximum temperature ($t_{max}$), mean air temperature at a height of 2 meters above the ground ($t_{mean}$), temperature amplitude ($t_{ampl}$), indicative of the temperature range between daytime high and nighttime low, mean relative humidity (*humidity*), mean water vapor pressure (*pressure*), average wind speed (*wind*), and precipitation (*precipitation*). We also incorporate three thermal comfort indices: Effective Temperature Taking Wind Velocity (*ETV*), Wind Chill Index (*VCI*) devised by [33], and the Discomfort Index (*DI*) introduced by [34]. These indices amalgamate various meteorological variables to provide a comprehensive assessment of thermal comfort.

## Districts

In view of the expansive nature of the complete dataset and the relative comparability of district-level outcomes, a strategic decision was made to focus on a subset of districts that would serve as a representative sample. This approach was guided by considerations of geographical localization and the unique climate conditions characteristic of each district.

To offer an insightful cross-section, we present the findings from the data analysis of four carefully chosen districts, each offering distinctive characteristics. The selection criteria

revolved around the geographical location and prevailing climate attributes. Specifically, we have included the results of two of Slovakia's largest cities, the district encompassing our university hospital, and a district situated at the heart of the country.

For administrative purposes, the larger cities are divided into multiple segments; however, for analytical coherency, we treat them as unified districts. The districts under scrutiny encompass: the capital city, Bratislava (BA), situated in the western region of the country; Košice (KE), positioned in the eastern part of Slovakia; Banská Bystrica (BB), located at the country's geographical midpoint; Martin (MT), situated in the northern region of Slovakia, encompassing the district of our university hospital.

## Data analysis

The data were explored and analysed within the R environment [35], version 4.0.5. This process was facilitated by several essential libraries, including roll [36], tsintermittent [37], rangerts [38], cutpointr [39] and pROC [40].

In order to make the research reproducible, an R Notebook was created for each of the studied districts. The resulting reports are included as S1 File.

For readers seeking insights into the statistical methodologies employed, basic information can be gleaned from [41]. For an elucidation of Croston's method, a seminal resource is provided by [42]. Readers less familiar with concepts like logistic regression and Generalized Linear Models may find [43] helpful. For information on Random Forest for Time Series, a comprehensive reference can be located in RFTS.

## Time series *Ext* of extremely high number of strokes

A new time series of extremely high number of strokes, denoted as *Ext*, was derived from the stroke counts time series. This process unfolded as follows: to generate *Ext*, the stroke counts time series was initially smoothed using a seven-day rolling sum. The resultant smoothed time series underwent detrending via a smoothing spline. Subsequently, the detrended, smoothed stroke counts time series was transformed into a binary time series. Specifically, each value surpassing the 90th percentile of past values was assigned 1, while the others were set to 0.

An alternative approach to constructing the extremely high stroke counts time series was briefly explored. Here, the extremeness threshold was fixed at the 90th percentile of the first 500 values. However, these results are not presented as they yielded poorer outcomes compared to the dynamically constructed extremes.

### EDA

The dataset underwent Exploratory Data Analysis (EDA) to unravel its inherent patterns and characteristics.

Time series of stroke counts were summarized through basic statistics, including mean, standard deviation, minimum, lower quartile, median, upper quartile, maximum values of stroke counts within a year, along with the percentage of days devoid of stroke patients and the total count of strokes. A Lag-plot was employed to visually capture the temporal dependence between present and past stroke count values, spanning different time shifts (lags). To visualize the potential association between stroke counts and lagged meteorological variables, a Lag Cross-plot was employed, incorporating the stroke counts against the lagged meteorological factor time series.

The same visualizations were constructed for the smoothed, detrended time series of counts, and the smoothed meteorological time series. The stroke counts time series were first

smoothed using a seven-day rolling sum and subsequently detrended through a smoothing spline. Meteorological time series underwent smoothing via a seven-day rolling sum.

To evaluate potential associations, the Cross Correlation Function (CCF) was computed between the *Ext* time series and the meteorological time series for each weather factor. Additionally, CCF was explored for the detrended rolling sum of counts and the rolling mean of the meteorological time series.

## Modelling study: Multivariate logistic regression and Random Forest for Time Series

A modelling study was performed on the full-length *Ext* time series, using the lagged *Ext* time series and weather time series as predictors. Two methods were used to model the association between the extremely high number of stroke time series and the meteorological factors time series: multivariate logistic regression model (Generalized Linear Model, GLM) and Random Forest Time Series (RFTS) machine learning algorithm. RFTS was used with the default settings.

The primary objective entailed identifying pivotal meteorological factors that serve as potent predictors of extremely high stroke counts. To achieve this, Akaike's Information Criterion (AIC) was harnessed within the GLM framework. Conversely, in the RFTS approach, meteorological factors were ranked based on their Importance. For RFTS, the Gini index served as the impurity measure, subsequently permuted to derive the Importance values.

Both methodologies shared a common model structure, elegantly expressed using the Wilkinson-Rogers notation:

$$
\begin{aligned}
Ext_t \sim \quad & Ext_{t-1} + Ext_{t-2} + Ext_{t-3} + \\
& t_{\max,\, t-1} + t_{\max,\, t-2} + t_{\max,\, t-3} + \\
& t_{\min,\, t-1} + t_{\min,\, t-2} + t_{\min,\, t-3} + \\
& t_{\mathrm{ampl},\, t-1} + t_{\mathrm{ampl},\, t-2} + t_{\mathrm{ampl},\, t-3} + \\
& t_{\mathrm{mean},\, t-1} + t_{\mathrm{mean},\, t-2} + t_{\mathrm{mean},\, t-3} + \\
& pressure_{t-1} + pressure_{t-2} + pressure_{t-3} + \\
& humidity_{t-1} + humidity_{t-2} + humidity_{t-3} + \\
& wind_{t-1} + wind_{t-2} + wind_{t-3} + \\
& precipitation_{t-1} + precipitation_{t-2} + precipitation_{t-3} + \\
& WCI_{t-1} + WCI_{t-2} + WCI_{t-3} + \\
& DI_{t-1} + DI_{t-2} + DI_{t-3} + \\
& ETV_{t-1} + ETV_{t-2} + ETV_{t-3} + \\
& day + week + month + year
\end{aligned}
\tag{1}
$$

## Forecasting study: GLM, RFTS and Croston's method

In addition to the modelling study we performed also a sequential forecasting study. This study encompassed the same two methodologies—GLM and RFTS—employed in the modelling study. Additionally, we introduced Croston's method into the forecasting study as a benchmark. Croston's method, a univariate approach, derives forecasts by leveraging historical

values of the time series itself. Croston's method was used with the value of the smoothing parameter set to 0.5, which corresponds to the balance of the recent and past values.

Sequential one-step-ahead forecasting study was performed as follows: 1) we leveraged the previously outlined construction of the *Ext* time series to generate the binary time series up to time point $t$. 2) The value of *Ext* at time $t$ served as the ground truth against which predictions from time $t - 1$ to time $t$ were compared.

Predictions were obtained through three different methods: a) multivariate logistic regression model (GLM), structured as depicted in Eq (1), b) RFTS, structured as depicted in Eq (1), c) Croston's method. The forecasting started at day 500.

The performance of the prediction was assessed by the Receiver Operating Characteristic (ROC) curve, and sensitivity and specificity optimizing the Youden index were considered optimal. The null hypothesis of equality of the Area under the ROC (AUC) for the three forecasting methods was tested by the bootstrap method in the pair-wise manner (and in paired mode); the resulting p-values were adjusted for multiple hypotheses testing by the Benjamini-Hochberg method.

This forecasting study not only sheds light on the predictive capabilities of the chosen methodologies but also underlines their relative performance against one another.

## Results

Results are shown for Bratislava, only. Findings for the other three districts can be found in S1 File.

### Exploration of time series

EDA was performed for the original time series of stroke counts; the smoothed, detrended stroke counts; and finally, for the time series *Ext* of extremely high number of strokes, i.e., for the smoothed, detrended, binarized stroke counts.

**Time series of stroke counts—Description.** The time series of stroke counts in BA, for years 2015 to 2019 are exhibited on Fig 2.

At around 9% of days of a year there were no ischaemic stroke patients in BA. The annual number of strokes in BA oscillated around 935 (the mean of the annual number of strokes over the five years); see Table 1.

The most frequent number of stroke counts was 2, except in the year 2016, when it was 3; as shown on Fig 3. For each year, the 90th percentile of stroke counts remained at 5.

**Time series of stroke counts—Autocorrelation.** As it can be seen on Fig 4, there is no or little auto-correlation in the stroke counts time series, regardless of lag.

**Time series of stroke counts—Cross-correlation with meteorological time series.** Fig 5 exhibits the lag cross-plot between the stroke counts on the $y$-axis and lagged values of the maximal daily temperature $t_{max}$ for lags from 0 (i.e., no lag) up to 5.

The lag cross-plot for the stroke counts vs. the maximal daily temperature suggests that there is a negligible association between the past values of the maximal daily temperature and the number of strokes. Similar conclusions hold for all the other meteorological variables.

To sum up, forecasting the next day number of stroke counts appears to be difficult, since there is a negligible association between the current and the past values of the stroke counts. Moreover, the meteorological variables cannot help in predicting the next value of stroke counts, as there is a negligible association between the stroke counts time series and the meteorological time series. This holds also for the other three districts, with desparate climatic conditions; see S1 File.

## Time series of stroke counts

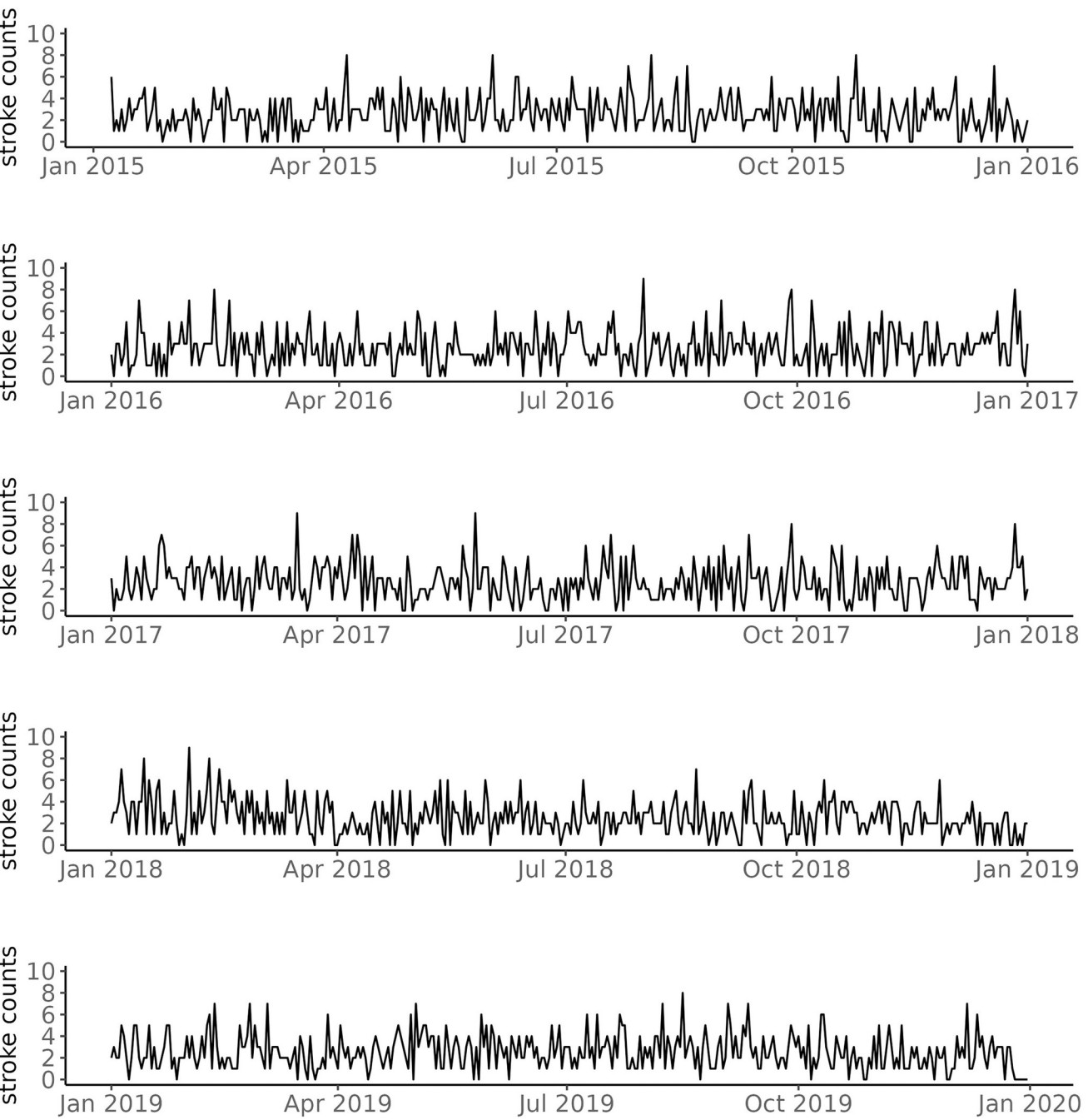

**Fig 2. Time series plot of stroke count time series for years 2015 to 2019, for BA.**

**Smoothed and detrended time series of stroke counts—Description.** The smoothed and detrended stroke counts time series are exhibited on Fig 6.

**Smoothed and detrended time series of stroke counts—Autocorrelation.** The seven day rolling-sum smoothing introduces an autocorrelation into the resulting smoothed time series

**Table 1. Descriptive statistics for the stroke counts in Bratislava, on yearly basis.** There, $n$ is the number of days in the year, for which the data are available; $sd$ is the standard deviation of the stroke counts, $Q1$, $Q3$ are the lower and upper quartiles, $percZero$ is the percentage of days with no stroke and $nstrokes$ is the number of strokes in the year; other abbreviations should be self-explanatory.

| year | n | mean | sd | min | Q1 | median | Q3 | max | percZero | nstrokes |
|------|-----|------|-----|-----|----|--------|----|-----|----------|----------|
| 2015 | 358 | 2.6 | 1.6 | 0 | 1 | 2 | 4 | 8 | 8.7 | 920 |
| 2016 | 366 | 2.6 | 1.7 | 0 | 1 | 2 | 3 | 9 | 9.6 | 949 |
| 2017 | 365 | 2.6 | 1.7 | 0 | 1 | 2 | 4 | 9 | 9.3 | 960 |
| 2018 | 365 | 2.5 | 1.6 | 0 | 1 | 2 | 3 | 9 | 9.0 | 912 |
| 2019 | 364 | 2.6 | 1.6 | 0 | 1 | 2 | 3 | 8 | 6.9 | 934 |

of stroke counts; see Fig 7. The strength of the autocorrelation decreases with the lag and at the lag 6 it becomes negligible.

**Smoothed and detrended time series of stroke counts—Cross-correlation with smoothed meteorological time series.** Lag cross-plot for the smoothed, detrended time series of stroke counts vs. the lagged rolling-mean detrended maximal daily temperature is exhibited on Fig 8. The cross-correlation is essentially non-existent.

This is confirmed by the Cross-Correlation Function (CCF) plot (see Fig 9), which provides a different view of the association between the smoothed, detrended time series of counts and the smoothed time series of meteorological variables, at different lags. Note that the cross correlation is computed between the smoothed, detrended stroke counts at time $t + i$ and the smoothed meteorological time series at time $t$, for $i = 0, \pm 1, \ldots, \pm 15$. The positive values of lag are of interest, here.

To sum up, the seven day roll-sum smoothing have introduced a moderate and short-term autocorrelation into the stroke counts time series. The seven day rolling mean smoothed meteorological time series exhibit negligible cross-correlation with the smoothed and detrended time series of stroke counts.

**Smoothed, detrended, binarized time series of stroke counts (*Ext*)—Description.** After smoothing and detrending the raw time series of stroke counts, the resulting time series were transformed into a time series of extremes (denoted *Ext*) by a binarization. To this end, the 90th percentile of the smoothed, detrended times series of stroke counts was computed and every value below the percentile cutoff was turned into 0; and every value above the cutoff was turned to 1. The resulting time series of extremely high number of strokes are depicted on Fig 10, for all the four districts.

**Smoothed, detrended, binarized time series of stroke counts (*Ext*)—Cross-correlation with smoothed meteorological time series.** The time series *Ext* of the extremely high number of stroke counts, which is just the roll-sum smoothed, detrended, binarized stroke counts time series, exhibit a small cross-correlation with some of the meteorological variables; see Fig 11.

For the positive values of lag which are of interest here, the cross correlations are at best around 0.05, ignoring its sign. Note that for majority of the weather variables the cross correlations with the *Ext* time series have increased, relative to those for the smoothed, detrended stroke counts. Also note that the sign of cross-correlation has changed for many of the meteorological factors. For instance, there was a positive cross correlation between the smoothed, detrended stroke counts and lag-one $t_{\max}$ (see, Fig 9A). For the time series *Ext* of the unusually high number of strokes the cross correlation with the lagged maximal daily temperature is negative, for every lag.

To sum up, the cross correlations of weather time series with the binary time series *Ext* are more pronounced than those for the smoothed, dentrended stroke counts. However, the size

## Barplot of stroke counts frequency

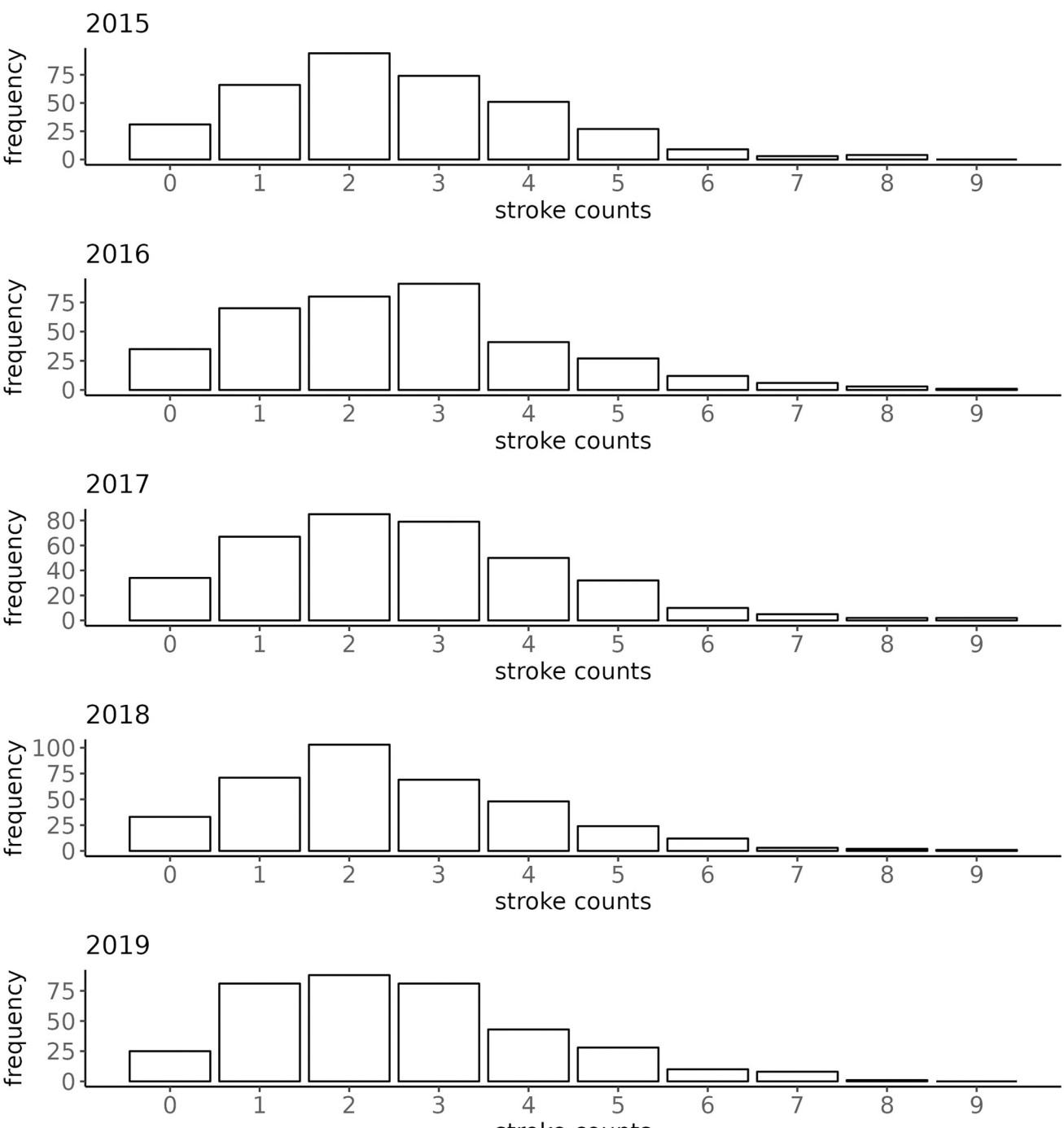

**Fig 3. Barplot of freqeuncy of stroke counts, year by year; for BA.**

of the cross correlation does not exceed 0.05, indicating that the meteorological time series bear little useful information for forecasting whether the next day will see a surge of number of strokes or it will be a day with regular number of strokes. This holds true not only for Bratislava, but also for the other three districts.

## Lag plot of stroke counts

**Fig 4. Lag-plot of time series of stroke counts for BA.** At a subplot, the number of strokes at time $t$ on the $y$-axis is exhibited against the number of strokes at time $t - i$, where the lag $i$ goes from 1 to 6.

## Lag cross plot for stroke counts and maximal daily temperature

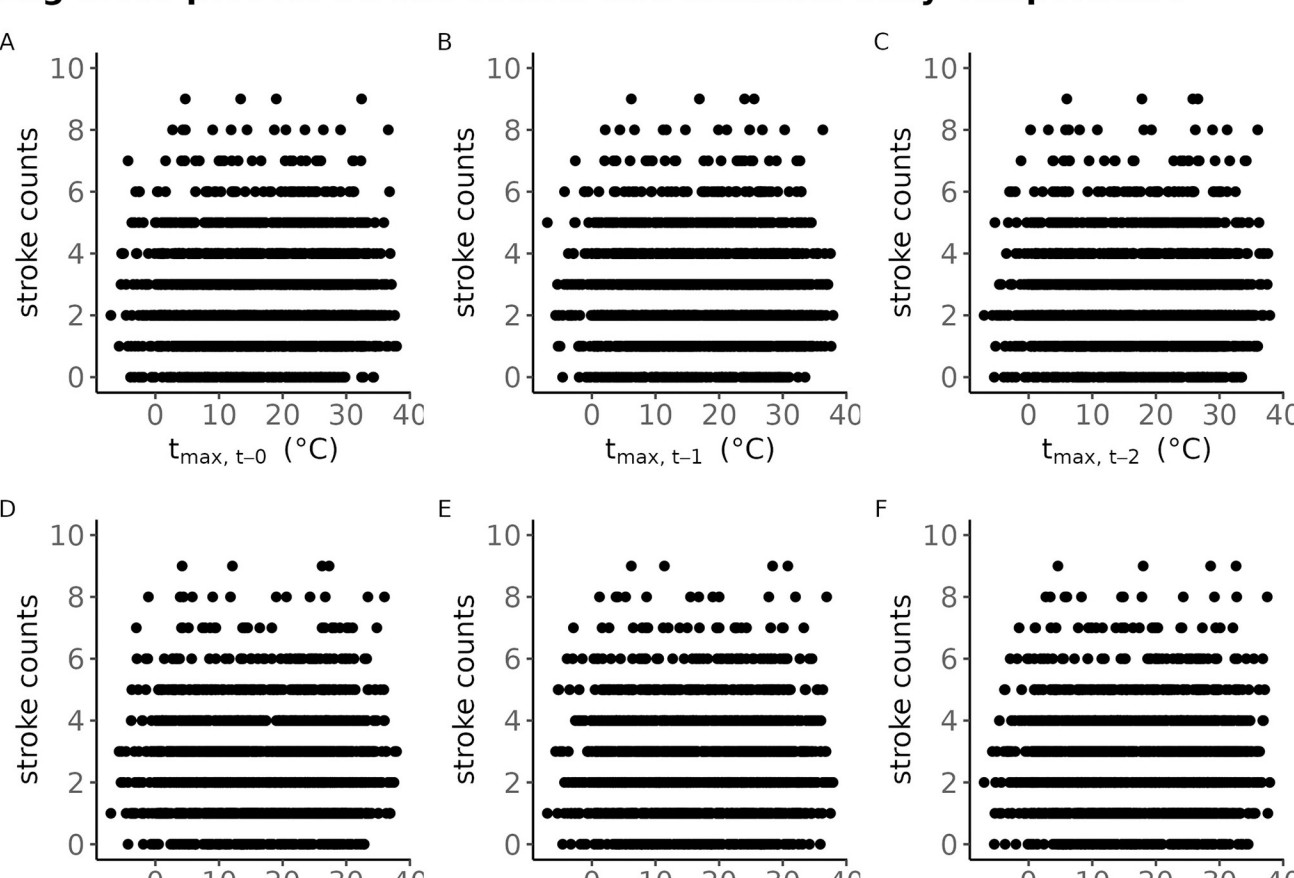

**Fig 5. Lag crossplot of time series of stroke counts vs. the lagged maximal daily temparuture; for BA.** At a subplot, the number of strokes at time $t$ on the $y$-axis is exhibited against the maximal daily temperature $t_{max}$ at time $t - i$, where the lag $i$ goes from 0 to 5.

### Multivariate logistic regression model of *Ext*

For Bratislava, the AIC simplified the full model (1) into the following one:

$$
\begin{aligned}
Ext_t \sim \quad & Ext_{t-1} + Ext_{t-3} + \\
& t_{\max,\, t-1} + t_{\max,\, t-2} + \\
& t_{\min,\, t-1} + t_{\min,\, t-2} + \\
& t_{\mathrm{ampl},\, t-1} + t_{\mathrm{ampl},\, t-2} + \\
& WCI_{t-2} + \\
& year
\end{aligned}
\tag{2}
$$

The forest plot of the standardized regression coefficients from the fitted model (2) is on Fig 12.

Due to the high uncertainty in the estimates for the temperature-related predictors, the forest plot makes it difficult to discern that the lagged values of *Ext* are the most important

## Time series of stroke counts, smoothed, detrended

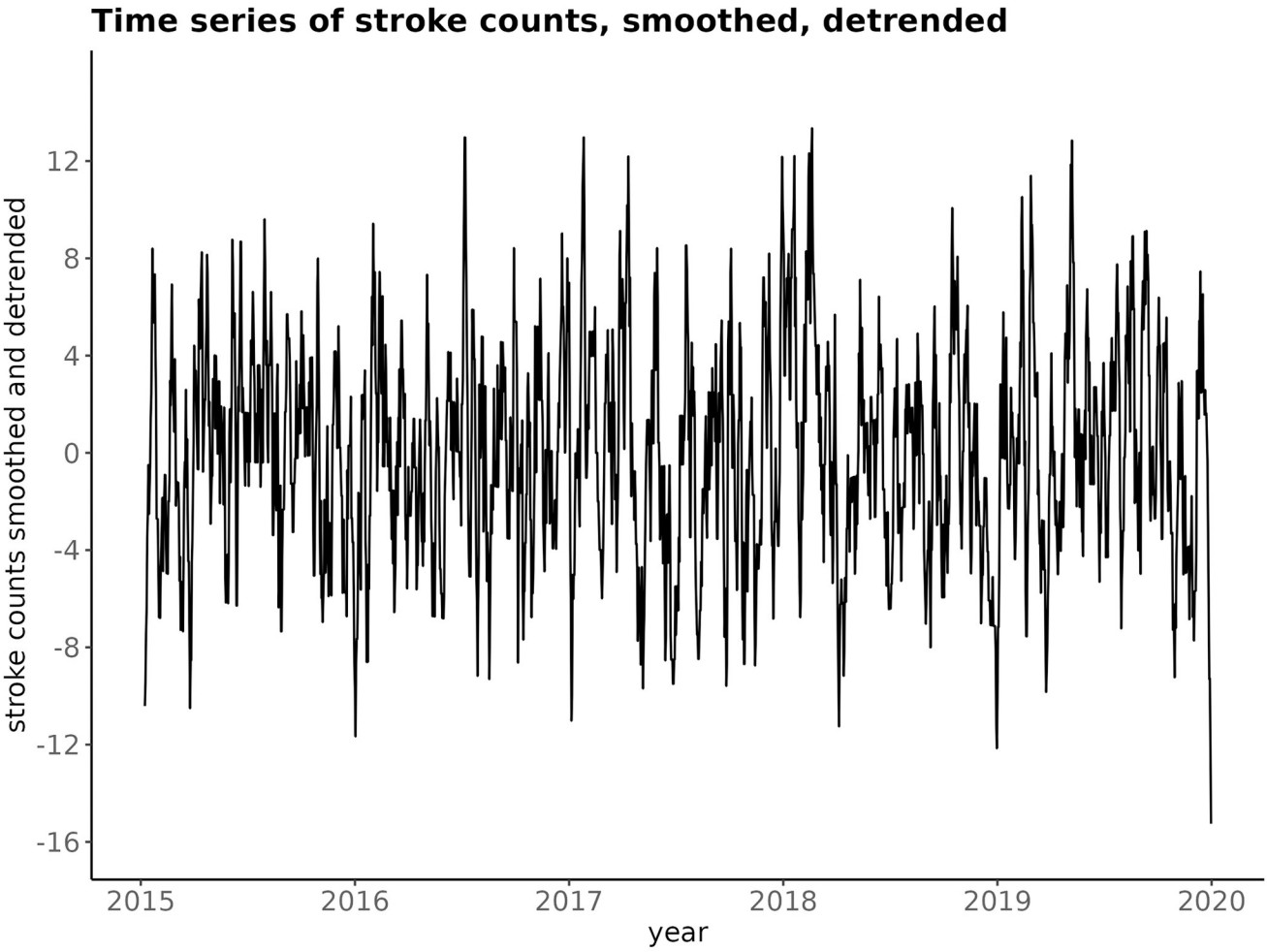

**Fig 6. Time series plot of the smoothed, detrended stroke counts time series, for BA.**

predictors of *Ext*. This can be concluded from Fig 13, where the forest plot for the subset of predictors is exhibited.

$Ext_{t-1}$ and $Ext_{t-3}$ are highly significant predictors of $Ext_i$; p-value $<0.01$ for the both predictors. For the meteorological variables selected by AIC the p-value oscillates around 0.05.

McFadden's pseudo-$R^2$ of the model is 0.32, which indicates a a good fit. The fit is due to primarily the past values of *Ext*, rather than the meteorological predictors. Indeed, the multi-variate logistic regression model with $Ext_1$, $Ext_3$ as the only predictors attains the McFadden's pseudo-$R^2$ = 0.30, which is worse only by 0.02. The Bayesian Information Criterion (BIC) even prefers the smaller model (BIC = 847) to the larger model (BIC = 897) which involves the meteorological variables.

Similar findings hold for the other three districts. At each location, some of the weather fac-tor time series are statistically significantly associated with the extremes time series. However, the most significant predictors of *Ext* are its past values.

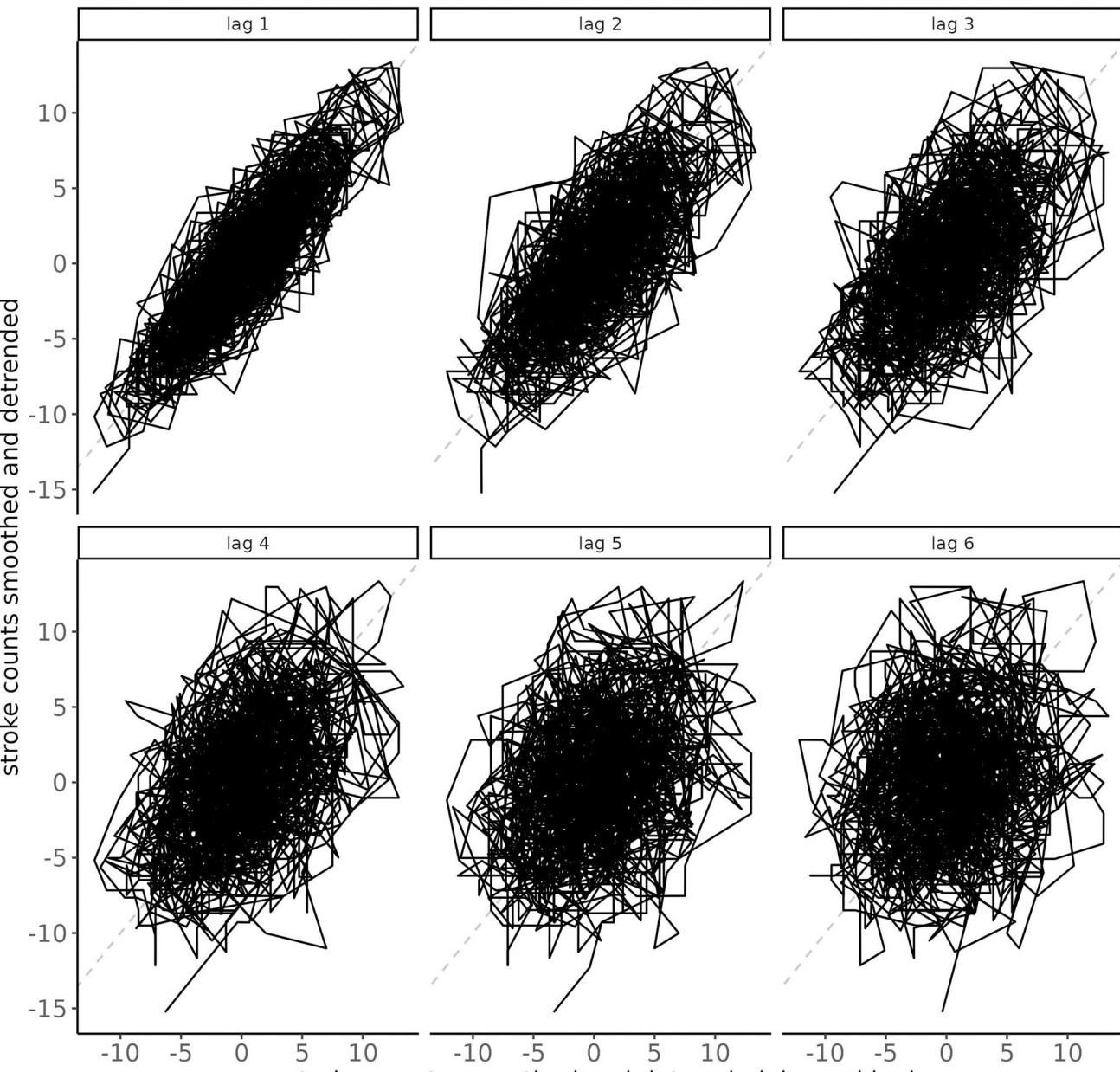

**Fig 7. Lag-plot of the smoothed and detrended time series of stroke counts for BA.** At a subplot, the smoothed, detrended number of strokes at time $t$ on the $y$-axis is exhibited against the smoothed detrended number of strokes at time $t - i$, where the lag $i$ goes from 1 to 6.

### Selection of important meteorological predictors of extreme stroke counts by Random Forest for Time Series

For BA, RandomForest for Time Series (RFTS) algorithm, with the same model (1) as that used in multivariate logistic regression, has identified the lag-1 extremes time series $Ext_{t-1}$ as the most important predictor. Its importance was almost twice as high as the importance of the second most important predictor—$DI$ at lag 1; see Fig 14.

## Lag crossplot for smoothed, detrended stroke counts and smoothed maximal daily temperature

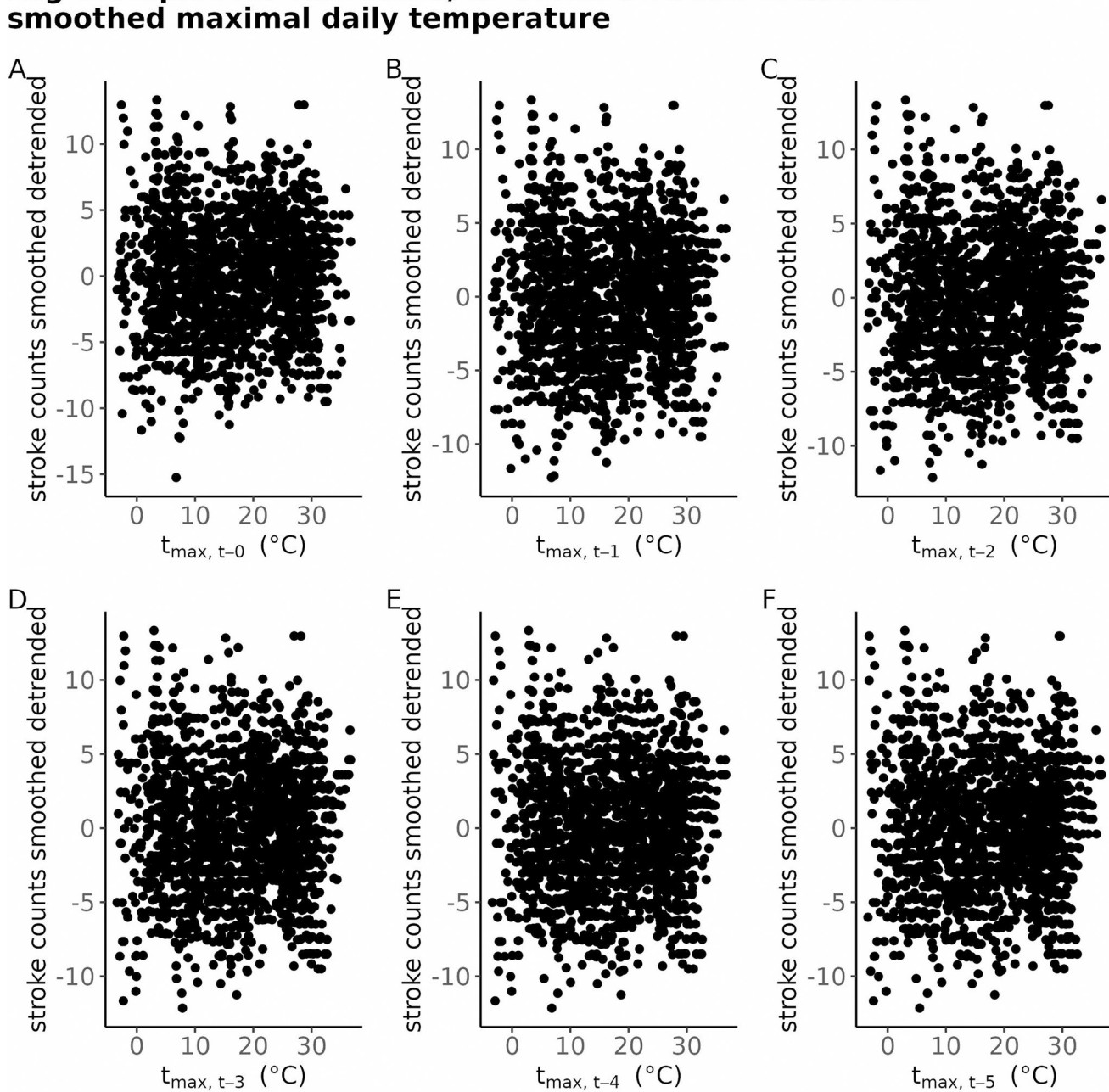

**Fig 8. Lag crossplot of the smoothed, detrended time series of stroke counts vs. the lagged smoothed maximal daily temparuture; for BA.** At a subplot, the smoothed and detrended number of strokes at time $t$ on the y-axis is exhibited against the smoothed maximal daily temperature $t_{max}$ at time $t - i$, where the lag $i$ goes from 0 to 5.

## CCF for smoothed, detrended stroke counts and weather time series

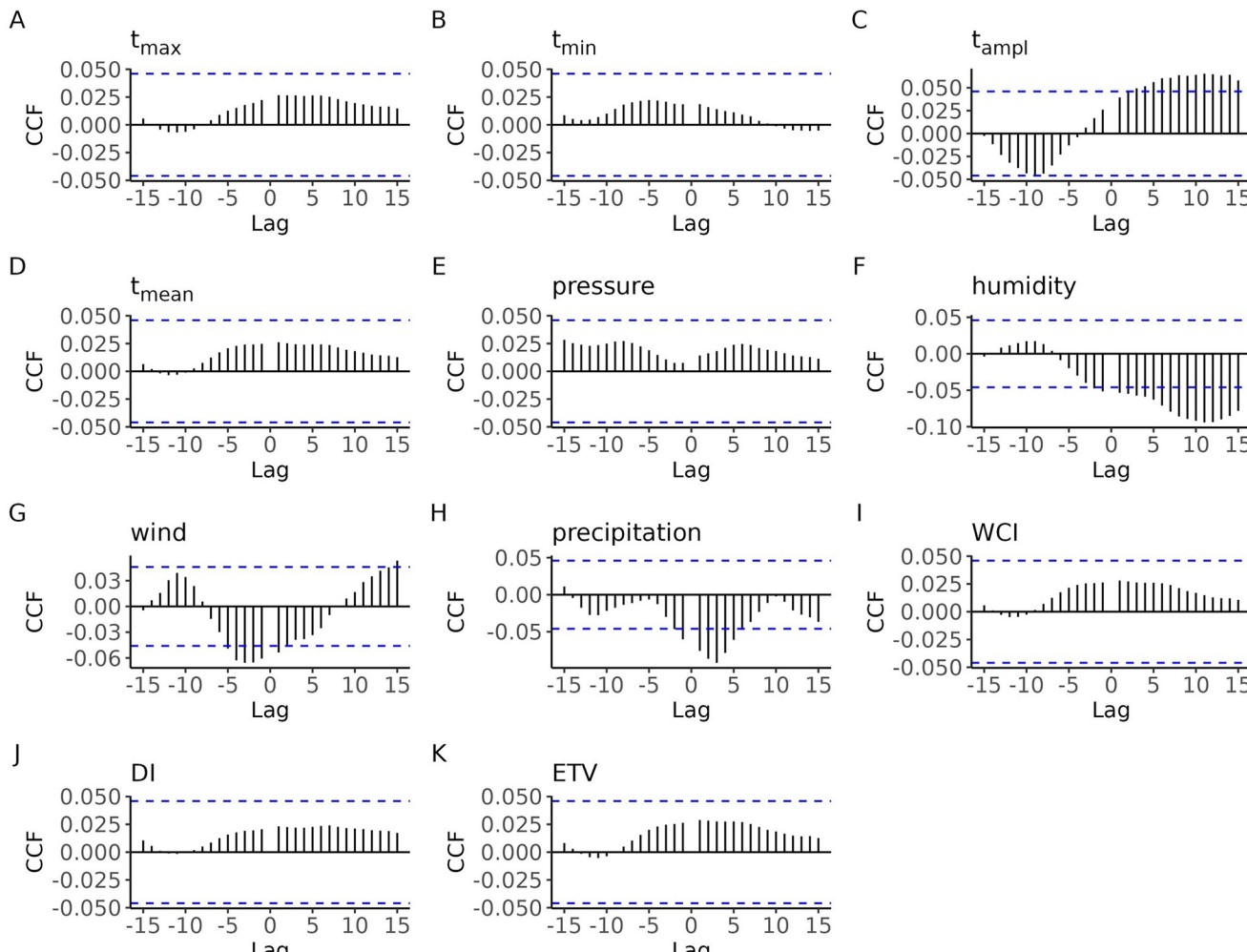

**Fig 9. Plot of CCF for the smoothed, detrended time series of stroke counts vs. the lagged smoothed meteorological time series; for BA.** The horizontal dashed lines depict the 95% confidence band for strict white noise.

Consistent with the ranking of predictors in multivariate logistic regression by statistical significance, the RFTS ranking selects $Ext_{t-1}$ as the single most important predictor. This holds for all four locations.

### Forecasting study

For Bratislava, the predictive accuracy, as measured by the Area under ROC (AUC), of the three studied forecasting methods was 0.64, 0.63, 0.56 for RFTS, GLM and Croston; respectively. AUC attained by RFTS was statistically significantly better than that of Croston method; see Table 2 for p-values and adjusted p-values for pairwise comparisons of AUC of all the three forecasting methods. Thus, AUC attained by the RFTS method which utilizes the meteorological time series was statistically significantly better than that of Croston's method, which bases its forecasts solely on the past values. Though the improvement in predictive performance due to the meteorological data was statistically significant, it was of a small magnitude.

## Time series of extremely high stroke counts (Ext)

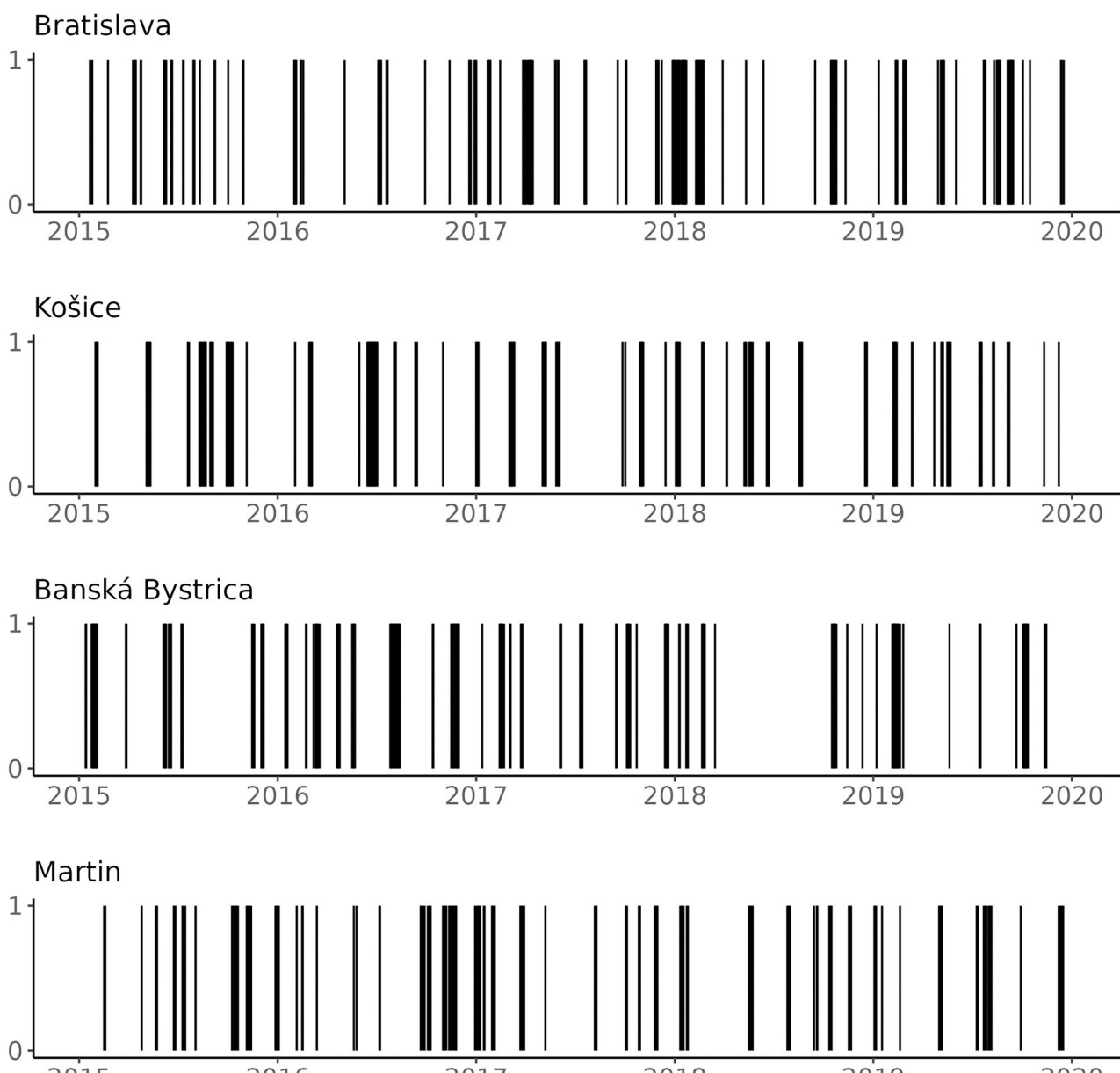

**Fig 10. Time series of extremely high number of strokes, for the four studied districts.**

ROC curves (see Fig 15) indicate that the sensitivity and specificity of forecasts is rather week. The best pair of values of sensitivity and specificity (0.626, 0.65) at Youden index cutoff is attained by GLM; see Table 3.

For the other three districts, detailed results of forecasting study can be found in S1 File. Values of AUC attained by the three methods in the three districts (KE, BB, MT) were around 0.6, similar to BA. Meteorological time series failed to improve predictive performance relative

## CCF for extremely high stroke counts and weather time series

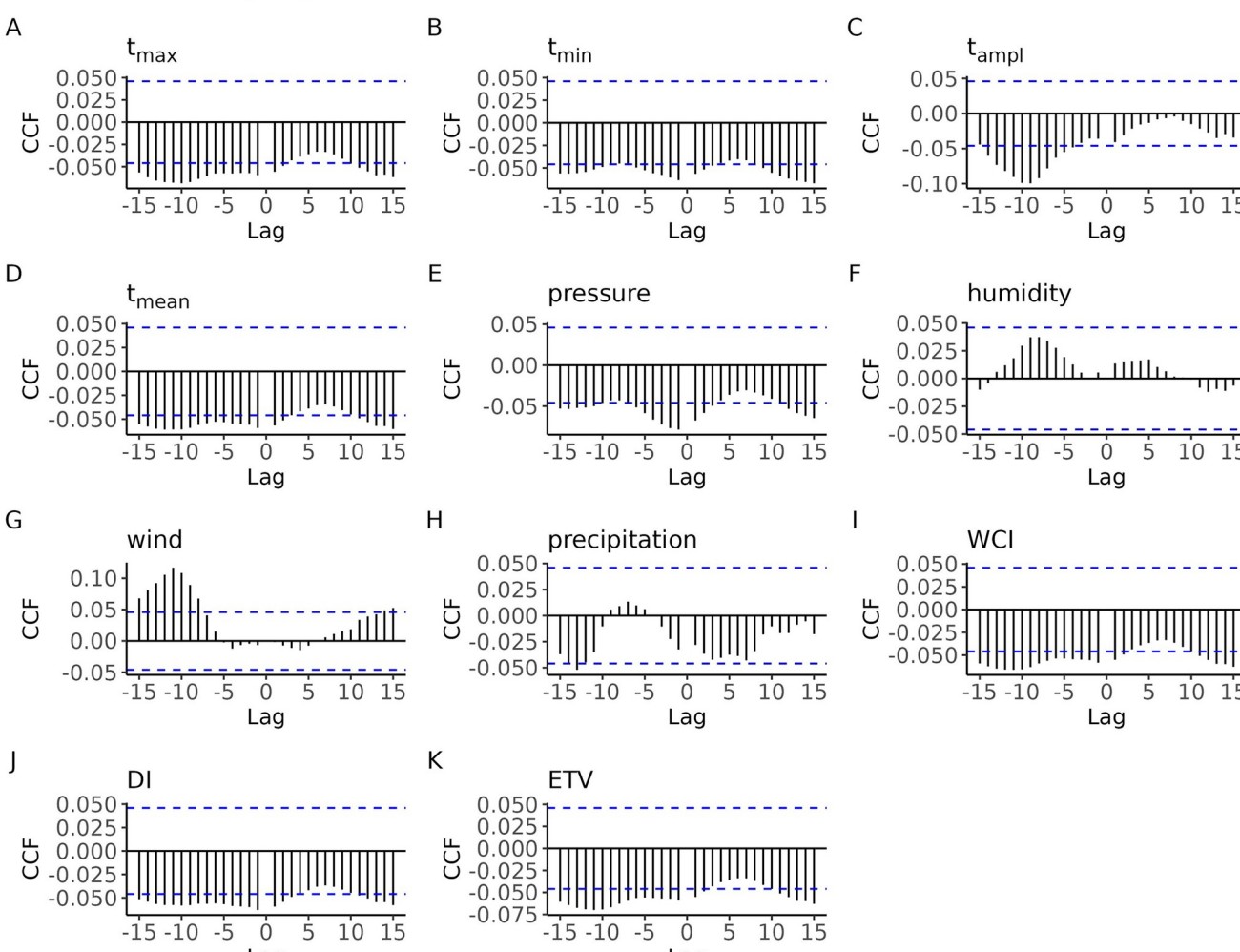

**Fig 11. Plot of CCF for the binary time series *Ext* of the unusually high number of strokes vs. the lagged smoothed meteorological time series; for BA.** The horizontal dashed lines depict the 95% confidence band for strict white noise.

to Croston's method in BB and MT. In KE, Croston's method attained statistically significantly better forecasting performance than RFTS and GLM.

## Discussion

In the context of ischaemic strokes and meteorological factors, the literature predominantly focuses on modeling associations between stroke count time series and meteorological time series. Typically, such endeavors reveal statistically significant links between certain meteorological factors and stroke counts. In our own study, we noted, for instance, that within the Bratislava district, the maximum daily temperature from the previous day exhibited a statistically significant association with the current values of the extremely high stroke counts time series.

However, the discourse on whether meteorological factors substantively augment stroke count forecasts and the extent of such improvement remains largely unaddressed in the existing literature. A noteworthy exception is the work by [22], who formulated a weather warning

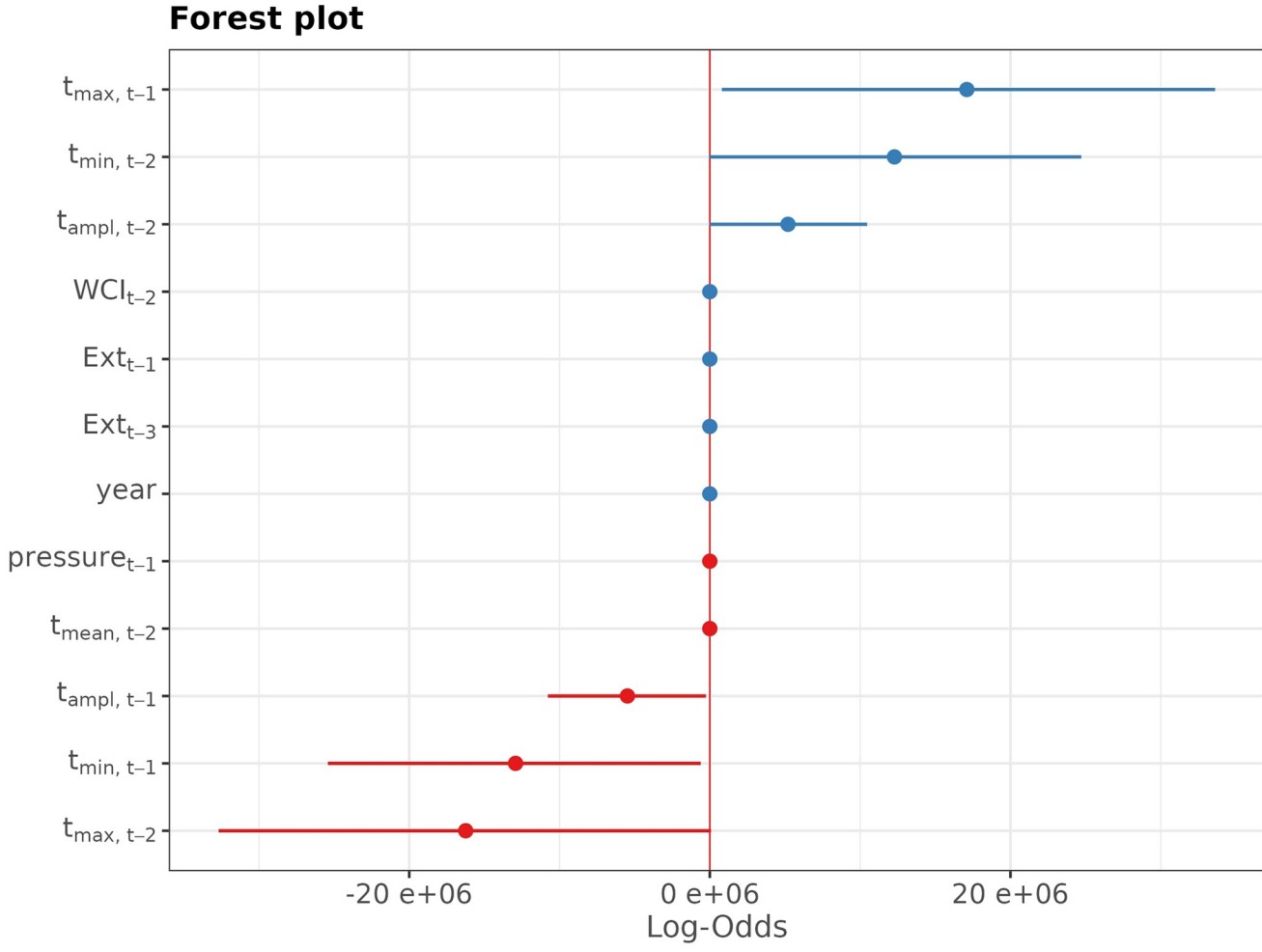

**Fig 12. Forest plot of the standardazied coefficients from the multivariate logistic regression model.** On the *x*-axis the value of the logarithm of odds-ratio (log-odds) is exhibitted for the predictors depicted at the *y*-axis. The horizontal line represents the 95% confidence band for the log-odds, which is depicted by the dot.

system for strokes based on South Korean meteorological data. They found that the predictive accuracy of their forecasts was limited.

Innovating within this landscape, our study stands as the first to probe the latent potential of meteorological factors in forecasting an extremely high number of ischaemic strokes. Through our investigation, we uncovered instances where meteorological factors indeed yield a statistically significant enhancement in forecasting extreme stroke counts, particularly within certain regions such as Bratislava. Nevertheless, it's vital to emphasize two critical aspects: *i)* the observed enhancement relative to the baseline Croston's method, whether significant or not, exhibited a minute magnitude; and *ii)* the baseline Croston's method, which relies solely on historical extreme stroke count values, yielded forecasts of notably low quality. Consequently, forecasting extreme stroke counts emerges as a formidable challenge, with the influence of meteorological factors offering limited assistance.

In addressing the limitations of our study, the insightful feedback from an anonymous reviewer and an Editor highlights the importance of considering air pollution alongside the meteorological variables we investigated, given its known association with stroke. Also,

## Forest plot (detail)

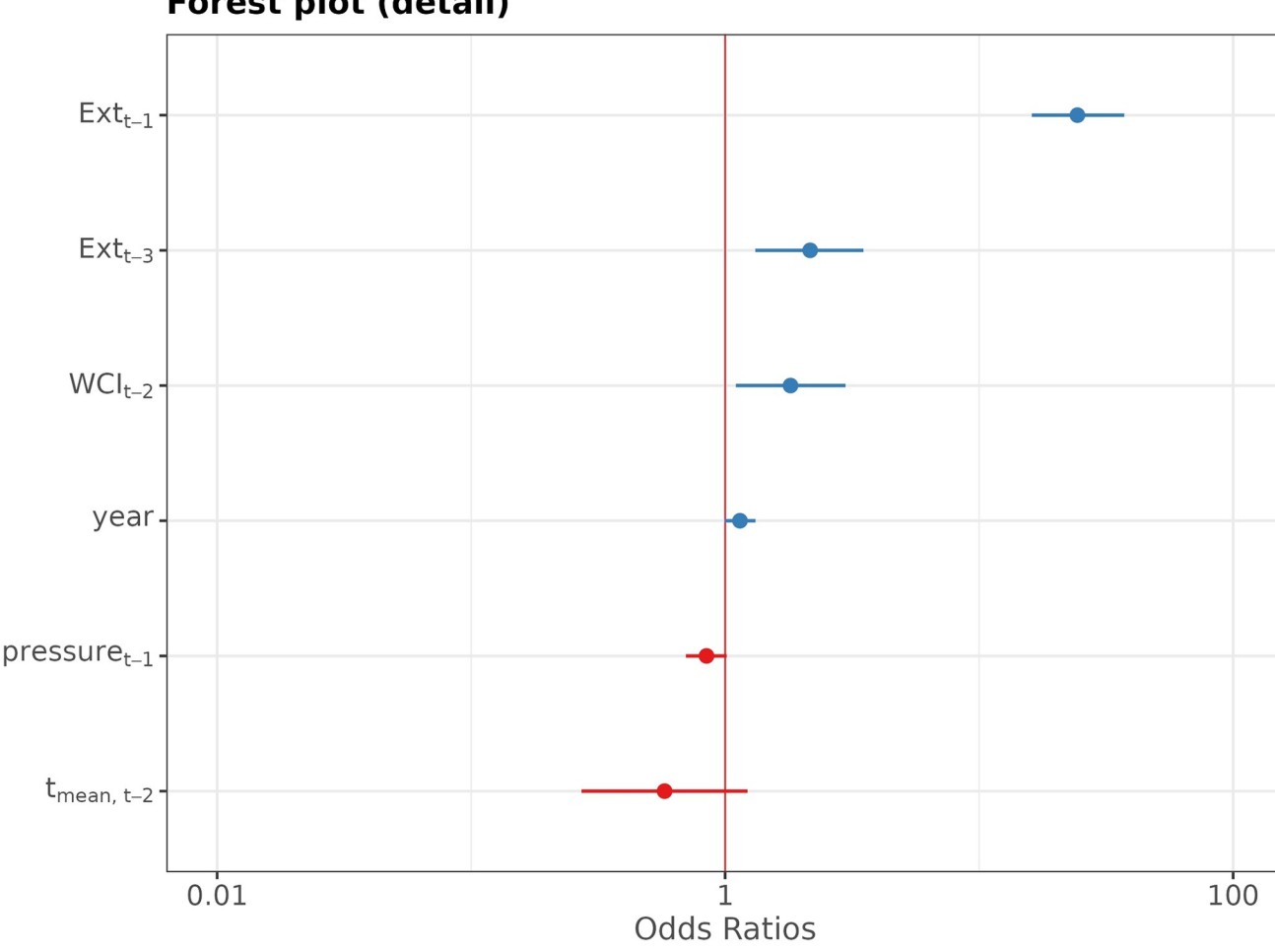

**Fig 13. Forest plot of the exponentially transformed coefficients from the multivariate logistic regression model.** On the *x*-axis the value of the odds-ratio (OR) is exhibited for the predictors depicted at the *y*-axis. The horizontal line represents the 95% confidence band for the OR, which is depicted by the dot.

socioeconomic status or healthcare access could affect stroke incidence. Furthermore, our approach to forecasting time series could be enhanced by incorporating spatiotemporal methods. Exploring alternative forecasting methods could also be valuable. Our study used Generalized Linear Regression, Random Forest for Time Series, and Croston's method as a starting point. We have made our data publicly available, inviting other researchers to test different approaches that may improve predictive accuracy. These valuable suggestions provide avenues for future research investigations.

Another reviewer suggested to add references to literature on using weather factors for predicting incidence of hospital admissions for other diagnoses. We searched for such a literature. We did it for cardiovascular disease. The state of art is very similar to that of ischaemic stroke —all the papers that we have scrutinized, model association between weather factors and the response (event, or number of events) and report which of the weather factors are significant, together with quantification of risk of the event. We were not able to find a research reporting a forecasting performance of the models. This reinforces the main points that we make by this research: *i)* many weather factors demonstrate statistically significant association with

## Variable importance by RFTS

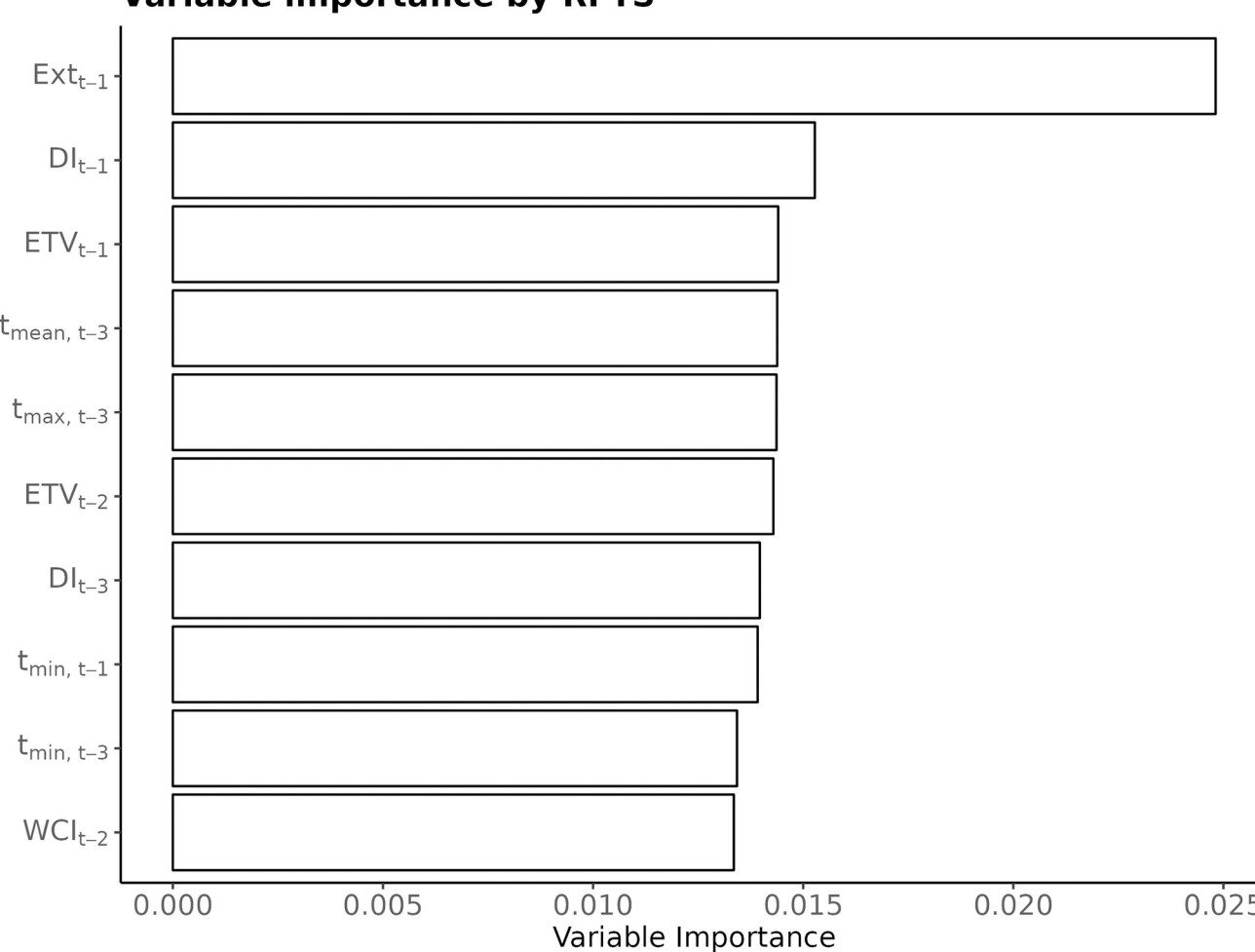

**Fig 14. RFTS importance plot of top-10 predictors.** Predictors are ranked along the *y*-axis, in the decreasing order. The value of Variable importance, depicted on *x*-axis indicates the relative importance of the predictor.

incidence of hospital admissions for a disease, *ii)* however, statistical significance not necessarily imply predictive/forecasting accuracy; *iii)* hence, it is of interest to assess the predictive power of the weather factors; *iv)* and to make it of practical interest, we decided to predict extremely high stroke counts so that adjustment of resources can be made to accommodate the predicted surge in stroke cases.

**Table 2. Comparing AUC attained by RFTS, GLM and Croston's method; for BA.** Adjusted p-values, denoted p-adj, were obtained by the Benjamini-Hochberg method.

| comparison | p-value | p-adj |
|---|---|---|
| RFTS vs GLM | 0.078 | 0.117 |
| RFTS vs Croston | 0.028 | 0.085 |
| GLM vs Croston | 0.324 | 0.324 |

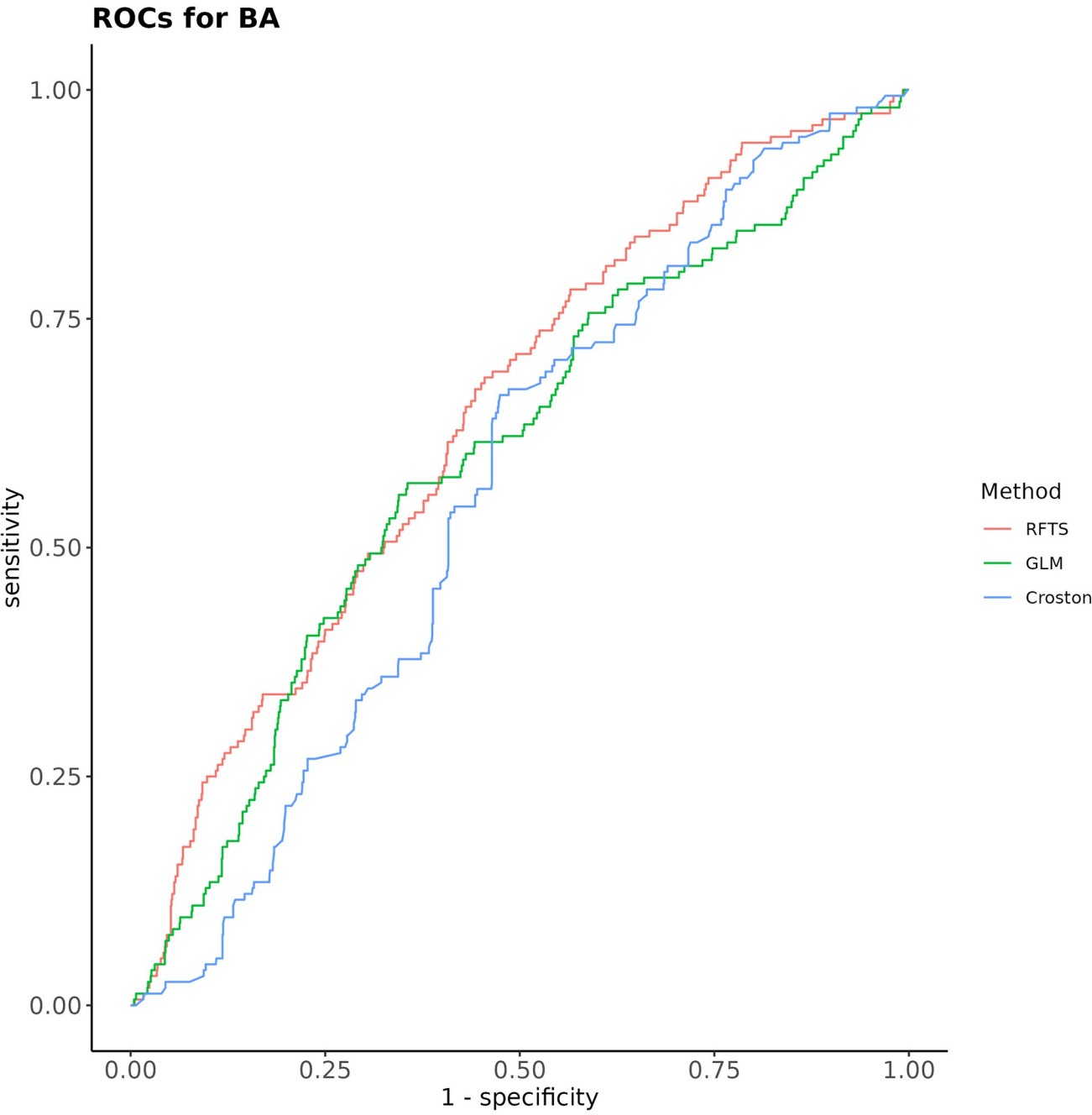

**Fig 15. ROCs attained by the three forecasting methods in the sequential forecasting study; for BA.**

**Table 3. Sensitivity and specificity at Youden index cutoff, for the three studied methods; district Bratislava.**

| method | sensitivity | specificity |
|--------|-------------|-------------|
| RFTS | 0.686 | 0.545 |
| GLM | 0.626 | 0.65 |
| Croston | 0.667 | 0.525 |

The reviewer also stressed that there are numerous features relevant to acute stroke [44], which should be used in forecasting. Unfortunately, the individual level data cannot be reconciled with the number of strokes data.

It is also worth stressing that our findings may be most relevant to regions with similar climatic conditions to Slovakia and may not be directly applicable to regions with significantly different meteorological conditions.

## Conclusions

While the connection between stroke incidence and meteorological conditions has spurred a multitude of studies, the realm of forecasting stroke incidence remains notably underexplored. Surprisingly little attention has been directed towards assessing the potential improvement in stroke incidence forecasting through the integration of meteorological information.

Our investigation, centered on the sequential forecasting of extremely high stroke counts, underscores several pivotal points. Firstly, across all three scrutinized forecasting methods (RFTS, GLM, Croston's method), forecasting accuracy remained consistently low. Secondly, among these methodologies, RFTS that harnessed meteorological factors showcased statistically significant advancements in forecasting accuracy within specific districts. However, it is imperative to underscore that this improvement, while statistically significant, is rather modest in its magnitude. Thirdly, there were districts where Croston's method attained significantly better forecasting performance than RFTS and/or GLM.

## Supporting information

**S1 File. Zip file with data and R source code.** The zip file contains RNotebook source code, data, and html report with detailed results of data analyses, for each district in a separate subdirectory. READ .ME file contains instructions for reproducing the results.
(ZIP)

## Acknowledgments

The authors are grateful to Martin Smatana, Ivana Loviskova and Martin Huba for help with stroke data completing and cleaning. Some of the suggestions by ChatGPT for improving English were incorporated into the final version of the manuscript.

## Author Contributions

**Conceptualization:** Lucia Babalova, Marian Grendar, Robert Mikulik, Vladimir Nosal.

**Data curation:** Lucia Babalova, Marian Grendar, Katarina Mikulova, Pavel Stastny, Pavel Fasko, Kristina Szaboova, Vladimir Nosal.

**Formal analysis:** Lucia Babalova, Marian Grendar, Egon Kurca, Stefan Sivak, Ema Kantorova, Peter Kubatka, Slavomir Nosal, Robert Mikulik, Vladimir Nosal.

**Investigation:** Marian Grendar, Vladimir Nosal.

**Methodology:** Lucia Babalova, Marian Grendar, Egon Kurca, Stefan Sivak, Ema Kantorova, Katarina Mikulova, Peter Kubatka, Slavomir Nosal, Robert Mikulik, Vladimir Nosal.

**Project administration:** Vladimir Nosal.

**Resources:** Vladimir Nosal.

**Supervision:** Robert Mikulik, Vladimir Nosal.

**Validation:** Lucia Babalova, Marian Grendar, Vladimir Nosal.

**Visualization:** Lucia Babalova, Marian Grendar, Katarina Mikulova, Pavel Stastny, Pavel Fasko, Kristina Szaboova, Vladimir Nosal.

**Writing – original draft:** Lucia Babalova, Marian Grendar.

**Writing – review & editing:** Lucia Babalova, Egon Kurca, Stefan Sivak, Ema Kantorova, Katarina Mikulova, Pavel Stastny, Pavel Fasko, Kristina Szaboova, Peter Kubatka, Slavomir Nosal, Robert Mikulik, Vladimir Nosal.

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
