## [Decision Letter · Decision Letter 0]

26 Dec 2023

PONE-D-23-38480Forecasting Extremely High Ischemic Stroke Incidence Using Meteorological Time SeriesPLOS ONE

Dear Dr. Nosal,

Thank you for submitting your manuscript to PLOS ONE. After careful consideration, we feel that it has merit but does not fully meet PLOS ONE’s publication criteria as it currently stands. Therefore, we invite you to submit a revised version of the manuscript that addresses the points raised during the review process.

ACADEMIC EDITOR: Major revision

We look forward to receiving your revised manuscript.

Kind regards,

Jyotir Moy Chatterjee, M. Tech (CSE)

Academic Editor

PLOS ONE

Journal Requirements:

5. We note that Figures 15 (SR_climate.jpg) and 16 (SR_station_district.jpg) in your submission contain [map/satellite] images which may be copyrighted. All PLOS content is published under the Creative Commons Attribution License (CC BY 4.0), which means that the manuscript, images, and Supporting Information files will be freely available online, and any third party is permitted to access, download, copy, distribute, and use these materials in any way, even commercially, with proper attribution. For these reasons, we cannot publish previously copyrighted maps or satellite images created using proprietary data, such as Google software (Google Maps, Street View, and Earth). For more information, see our copyright guidelines: http://journals.plos.org/plosone/s/licenses-and-copyright.

     1. You may seek permission from the original copyright holder of Figures 15 (SR_climate.jpg) and 16 (SR_station_district.jpg) to publish the content specifically under the CC BY 4.0 license.  

Reviewers' comments:

Reviewer's Responses to Questions

**Comments to the Author**

1. Is the manuscript technically sound, and do the data support the conclusions?

Reviewer #1: Partly

Reviewer #2: No

2. Has the statistical analysis been performed appropriately and rigorously? 

Reviewer #1: I Don't Know

Reviewer #2: No

3. Have the authors made all data underlying the findings in their manuscript fully available?

Reviewer #1: No

Reviewer #2: No

4. Is the manuscript presented in an intelligible fashion and written in standard English?

Reviewer #1: No

Reviewer #2: No

5. Review Comments to the Author

Reviewer #1: Figure no. information is missing to infer the flow of article. ?? in place on figure no. great hindrance. At some place mentioned not much impact between ischaemic stroke and meteorology parameter. Mention the purpose and optimize the map sort of figure given towards last.

Reviewer #2: The authors have used a number of ML methods to Forecast

Extremely High Ischemic Stroke Incidence Using Meteorological Time

Series

However there are some issues related to the paper.

There is no information about tuning the models, training and testing sets, uncertainties.

Also few meteorological data were used related to the Extremely High Ischemic Stroke Incidence. Extremely High Ischemic Stroke Incidence rate is also associated with air polution parameters which were ignored in this study.

Also it is not known why did the authors considered a binary response and have used logistic regression. They can use count response which is more informative. Age and sex were not considered. One thisg is that the used methods do not consider the spatial variations. According to the manuscript the authors have gathered the data from several regions therefore using spatio temporal would be more sound.

6. PLOS authors have the option to publish the peer review history of their article (what does this mean?). If published, this will include your full peer review and any attached files.

Reviewer #1: **Yes: **Sujatha Radhakrishnan

Reviewer #2: No

---

## [Author Response · Author response to Decision Letter 0]

17 Jan 2024

Response to Reviewers

Reviewer #1

Thank you for noting that figure numbering was missing in the manuscript. This was, in part, due to an error in Overleaf, which we have used for preparing the manuscript. It is corrected in the revised version.

Reviewer #2

The authors have used a number of ML methods to Forecast Extremely High Ischemic Stroke Incidence Using Meteorological Time Series

However there are some issues related to the paper.

There is no information about tuning the models, training and testing sets, uncertainties.

Also few meteorological data were used related to the Extremely High Ischemic Stroke Incidence. Extremely High Ischemic Stroke Incidence rate is also associated with air polution parameters which were ignored in this study.

Also it is not known why did the authors considered a binary response and have used logistic regression. They can use count response which is more informative. Age and sex were not considered. One thisg is that the used methods do not consider the spatial variations. According to the manuscript the authors have gathered the data from several regions therefore using spatio temporal would be more sound.

Thank you very much for the constructive criticism of our study! Below, we address yours question, and comments point by point:

- There is no information about tuning the models, training and testing sets, uncertainties.

Thank you for making this point!

RandomForest for Time Series was used with the default settings. There is no parameter to be tuned in Multivariate Logistic Regression. Croston’s forecasting method was used with the smoothing parameter w set to 0.5, which corresponds to the balance of the recent and past values.

We have added the following sentences to the text: 1) "RFTS was used with the default settings.", placed at the end of the first paragraph of Section 'Modelling study: Multivariate logistic regression and Random Forest for Time Series. 2) "Croston's method was used with the value of the smoothing parameter set to 0.5, which corresponds to the balance of the recent and past values.", placed at the end of the first paragraph of Section 'Forecasting study: GLM, RFTS and Croston’s method'.

As far as the training and testing is concerned, as we state in the paper (see Forecasting study: GLM, RFTS and Croston’s method), the time series of first 500 observations was used for the initial training. The forecasts were sequentially updated. We were interested in point forecasts and have quantified the quality of forecasts by the ROC curve.

- Also few meteorological data were used related to the Extremely High Ischemic Stroke Incidence. Extremely High Ischemic Stroke Incidence rate is also associated with air polution parameters which were ignored in this study.

Thank you for pointing out that other meteorological variables should also be considered. Unfortunately, we did not have access to the air pollution data.

- Also it is not known why did the authors considered a binary response and have used logistic regression. They can use count response which is more informative.

Thank you for the comments! Our motivation for turning the counts time series into binary time series is explained in the Introduction of the original submission, where we state:

"In the pursuit of making our approach clinically relevant, we chose to forecast extremely high stroke counts. This facet holds practical significance, enabling the adjustment of resources to accommodate the predicted surge in stroke cases. Notably, forecasting high stroke counts sidesteps the challenge of selecting a quality measure for predictions. By transforming raw stroke counts into binary time series (ordinary stroke count/high stroke count), we can employ sensitivity and specificity metrics to quantify forecast quality."

- Age and sex were not considered.

We are not aware of any method for relating together the scalar number of strokes and a vector of gender (or age) of the patients who had the stroke. To fix the ideas, let at day d and district D there be k patients with stroke. Sex of the patients is a vector of length k; say, (Female, Female, Male, Male, Male) assuming that k = 5. How could be the vector of gender used to predict the number of strokes, which is a scalar? The vector should first be turned into a scalar. But how? By averaging it? The other option would be to add gender vector as the k predictors. But, at each day there would be different number of columns in the design matrix. This is not viable, either.

- One thisg is that the used methods do not consider the spatial variations. According to the manuscript the authors have gathered the data from several regions therefore using spatio temporal would be more sound.

Again, many thanks for the suggestion! Spatiotemporal modeling would definitely be more encompassing, as the forecasts in one place could borrow strength from data in the other districts. This could be an interesting venue to follow in the future research. We have added it to the limitations of the study discussion.

The following paragraph was added to the Discussion:

In addressing the limitations of our study, the insightful feedback from an anonymous reviewer highlights the importance of considering air pollution alongside the meteorological variables we investigated, given its known association with stroke. Furthermore, our approach to forecasting time series could be enhanced by incorporating spatiotemporal methods. These valuable suggestions provide avenues for future research investigations.

---

## [Decision Letter · Decision Letter 1]

8 Apr 2024

PONE-D-23-38480R1Forecasting extremely high ischemic stroke incidence using meteorological time seriePLOS ONE

Dear Dr. Nosal,

Thank you for submitting your manuscript to PLOS ONE. After careful consideration, we feel that it has merit but does not fully meet PLOS ONE’s publication criteria as it currently stands. Therefore, we invite you to submit a revised version of the manuscript that addresses the points raised during the review process.

**Minor Revision**

We look forward to receiving your revised manuscript.

Kind regards,

Jyotir Moy Chatterjee

Academic Editor

PLOS ONE

Journal Requirements:

Reviewers' comments:

Reviewer's Responses to Questions

**Comments to the Author**

1. If the authors have adequately addressed your comments raised in a previous round of review and you feel that this manuscript is now acceptable for publication, you may indicate that here to bypass the “Comments to the Author” section, enter your conflict of interest statement in the “Confidential to Editor” section, and submit your "Accept" recommendation.

Reviewer #3: All comments have been addressed

Reviewer #4: (No Response)

2. Is the manuscript technically sound, and do the data support the conclusions?

Reviewer #3: Partly

Reviewer #4: Yes

3. Has the statistical analysis been performed appropriately and rigorously? 

Reviewer #3: I Don't Know

Reviewer #4: Yes

4. Have the authors made all data underlying the findings in their manuscript fully available?

Reviewer #3: Yes

Reviewer #4: Yes

5. Is the manuscript presented in an intelligible fashion and written in standard English?

Reviewer #3: Yes

Reviewer #4: Yes

6. Review Comments to the Author

**Reviewer #3**: Authors addressed all the comments and concerned carefully. The manuscript stands for the acceptance.

**Reviewer #4:** This appears to be a well performed and reported study. However, in our mind the results are rather expected. That is, given the small number of high stroke counts, and the small number of strokes even in the extreme cases, and additionally given the very many factors affecting stroke - it is rather unlikely that the models employed produce accurate predictions. It would be advisable to add references to literature that shows the limited capability in such cases (even if not for stroke prediction), and to the numerous features relevant to acute stroke (e.g., Mazza et al. 2021). Possibly, a much larger dataset could have helped.

7. PLOS authors have the option to publish the peer review history of their article (what does this mean?). If published, this will include your full peer review and any attached files.

Reviewer #3: No

Reviewer #4: No

---

## [Author Response · Author response to Decision Letter 1]

9 Apr 2024

Dear Jyotir Moy Chatterjee, M. Tech (CSE),

thank you, again, for the care you gave to considering our manuscript Forecasting extremely high ischemic stroke incidence using meteorological time series and for selecting knowledgeable reviewers!

We have addressed the comments and suggestions in the revised version of the manuscript. Detailed response to reviewers can be found at the bottom of this document.

Concerning the Journal requirements:

1) We have reviewed the reference list for its completeness and correctness. Upon a suggestion from Reviewer #4 we have added another reference (#44) to the References. The change is marked also in the Revised Manuscript with Track Changes pdf file. 

Thank you, again, for yours editorial work!

Best regards,

 Vladimir Nosal, PhD, MD.

 Martin, Apr 9, 2024

Response to Reviewers

Reviewer #4

Prompted by your suggestion to add references that show the limited capability of weather factors to forecast incidence of hospital admissions for other diseases, we searched for such a literature. We did it for cardiovascular disease. The state of art is very similar to that of ischaemic stroke - all the papers that we have scrutinized, model association between weather factors and the response (event, or number of events) and report which of the weather factors are significant, together with quantification of risk of the event. We were not able to find a research reporting a forecasting performance of the models. This reinforces the main points that we make by this research:

1) many weather factors demonstrate statistically significant association with incidence of hospital admissions for a disease,

2) however, statistical significance not necessarily imply predictive/forecasting accuracy;

3) hence, it is of interest to assess the predictive power of the weather factors;

4) and to make it of practical interest, we decided to predict extremely high stroke counts so that adjustment of resources can be made to accommodate the predicted surge in stroke cases.

We add the above text to the Discussion section of the revised manuscript.

We agree with your point that there are numerous features relevant to acute stroke and we add the reference you have suggested to the manuscript. On technical level, however, it should be noted that the patient level data (such as gene expression, age, gender, etc.) cannot be reconciled with the counts data. 

Thank you for your comments and suggestions!

---

## [Decision Letter · Decision Letter 2]

9 Aug 2024

PONE-D-23-38480R2Forecasting extremely high ischemic stroke incidence using meteorological time seriesPLOS ONE

Dear Dr. Nosal,

Thank you for submitting your manuscript to PLOS ONE. After careful consideration, we feel that it has merit but does not fully meet PLOS ONE’s publication criteria as it currently stands. Therefore, we invite you to submit a revised version of the manuscript that addresses the points raised during the review process.

**ACADEMIC EDITOR: Minor Revision** For Lab, Study and Registered Report Protocols: These article types are not expected to include results but may include pilot data. 

We look forward to receiving your revised manuscript.

Kind regards,

Jyotir Moy Chatterjee

Academic Editor

PLOS ONE

Journal Requirements:

Additional Editor Comments:

The work outlined a comprehensive study on the relationship between weather conditions and ischemic stroke incidence, focusing on predictive modeling. Below are 20 potential limitations of the work:

1. Keywords are missing.

2. The main contribution of the work is not clear from the introduction.

3. The overall organization of the paper is not presented in the introduction section.

4. The study is based on data from stroke centers in the Slovak Republic, which may not be representative of other regions with different climates or healthcare systems.

5. The five-year period of data collection might not capture long-term trends or rare events that could affect stroke incidence.

6. Findings may not be applicable to populations outside the Slovak Republic, particularly those in regions with significantly different meteorological conditions.

7. The quality and consistency of the meteorological and stroke data across different centers might vary, potentially introducing bias.

8. The study may not adequately account for other factors influencing stroke incidence, such as air pollution, socioeconomic status, or healthcare access.

9. By concentrating on extreme stroke events (90th percentile), the study may overlook insights into more common stroke occurrences, which could provide a broader understanding.

10. The transformation of stroke counts into a binary outcome (high vs. ordinary) might oversimplify the complex nature of stroke incidence, potentially missing subtler trends.

11. The limited predictive accuracy of all three methods suggests that the models may not be well-suited for forecasting extreme stroke events, indicating a need for alternative approaches.

12. Croston’s method, which relies solely on historical stroke data, may not capture the influence of changing meteorological conditions over time.

13. The use of multivariate logistic regression and Random Forest models could lead to overfitting, particularly if the models are overly complex relative to the size of the dataset.

14. The study may not have identified the most relevant meteorological variables influencing stroke incidence, leading to weak correlations.

15. The daily resolution of meteorological data might be insufficient to capture finer-scale weather patterns that could impact stroke risk.

16. The study does not appear to consider potential temporal lags between changes in weather conditions and stroke incidence, which could affect the accuracy of the predictions.

17. The complexity of Random Forest and logistic regression models may make it difficult to interpret the relationship between specific meteorological factors and stroke incidence.

18. The models were developed and tested on the same dataset, which may limit the validity of the findings when applied to other datasets or populations.

19. While the study discusses seasonality, the forecasting models do not explicitly incorporate seasonal patterns, which could be a significant factor in stroke incidence.

20. The study focuses on meteorological factors and may overlook other critical factors, such as lifestyle changes, that could influence stroke incidence.

21. The number of extreme stroke events (surpassing the 90th percentile) may be small, leading to challenges in building robust predictive models.

22. The binary prediction approach may result in a high rate of false positives or false negatives, reducing the practical utility of the forecasts.

23. Using predictive models to forecast extreme medical events could raise ethical concerns, particularly if the forecasts are not accurate enough to inform effective interventions.

Reviewers' comments:

Reviewer's Responses to Questions

**Comments to the Author**

1. If the authors have adequately addressed your comments raised in a previous round of review and you feel that this manuscript is now acceptable for publication, you may indicate that here to bypass the “Comments to the Author” section, enter your conflict of interest statement in the “Confidential to Editor” section, and submit your "Accept" recommendation.

Reviewer #4: All comments have been addressed

2. Is the manuscript technically sound, and do the data support the conclusions?

Reviewer #4: Partly

3. Has the statistical analysis been performed appropriately and rigorously? 

Reviewer #4: Yes

4. Have the authors made all data underlying the findings in their manuscript fully available?

Reviewer #4: Yes

5. Is the manuscript presented in an intelligible fashion and written in standard English?

Reviewer #4: Yes

6. Review Comments to the Author

Reviewer #4: The addition to the discussion addresses most of my concerns. However the new reference (#44) is not the right reference - it is mostly irrelevant. It happens to have the same author name (i.e., Mazza), but it is not about acute stroke and the associated features.

Try :Machine Learning Techniques in Blood Pressure Management During the Acute Phase of Ischemic Stroke by Mazza et al. instead, which is the relevant one.

7. PLOS authors have the option to publish the peer review history of their article (what does this mean?). If published, this will include your full peer review and any attached files.

Reviewer #4: No

---

## [Author Response · Author response to Decision Letter 2]

11 Aug 2024

Dear Jyotir Moy Chatterjee, M. Tech (CSE),

thank you, again, for the care you gave to considering our manuscript Forecasting extremely high ischemic stroke incidence using meteorological time series and for yours comments and suggestions!

We have addressed the comments and suggestions in the revised version of the manuscript. Detailed response to the Additional Editor Comments can be found at the bottom of this document.

Concerning the Journal requirements:

1) We have reviewed the reference list for its completeness and correctness. Upon a suggestion from Reviewer #4 we have replaced reference (#44) by the one suggested by the Reviewer. The change is marked also in the Revised Manuscript with Track Changes pdf file. 

Thank you, again, for yours editorial work!

Best regards,

 Vladimir Nosal, PhD, MD.

Martin, Aug 10, 2024

Response to Additional Editor Comments

Thank you for your insightful comments and suggestions regarding the limitations of our study. We greatly appreciate the time and effort you put into reviewing our manuscript.

1. Keywords are missing.

Thank you for pointing this out. We checked the official PLoS ONE LaTeX template (https://journals.plos.org/plosone/s/latex) and noted that there is no option available for providing keywords. We also checked a couple recently published papers from PLoS ONE and no keywords are presented in the papers.

(https://journals.plos.org/plosone/article?id=10.1371/journal.pone.0307842

https://journals.plos.org/plosone/article?id=10.1371/journal.pone.0305825)

However, if keywords are needed for other purposes, we suggest the following: ischemic stroke; weather; time series forecasting;

2. The main contribution of the work is not clear from the introduction.

Thank you for pointing out that the main contribution of the work is not stated clearly.

Original formulation of the main contribution:

"We propose an augmentation of traditional association modeling through stroke count forecasting,

intending to achieve the following objectives:

i) Evaluate the predictive efficacy of different forecasting methods for stroke counts;

ii) Compare forecasts leveraging meteorological time series against those grounded solely in historical stroke count data."

was in the revised version reformulated into a more impactful form:

"With this perspective, in our study, we augment traditional association modeling by incorporating stroke count forecasting. Our primary contribution is twofold:

Evaluation of Predictive Methods: We systematically evaluate the predictive efficacy of different forecasting methods, including Generalized Linear Regression (GLM), RandomForest for Time Series, and Croston's method, in predicting stroke counts.

Comparison of Forecasting Approaches: We compare forecasts that leverage meteorological time series data against those based solely on historical stroke count data. This comparison aims to determine whether incorporating meteorological data improves the accuracy and utility of stroke count predictions."

3. The overall organization of the paper is not presented in the introduction section.

Thank you for making the point! We checked the above-mentioned two papers recently published by PLoS ONE, and they do not contain paper organization in the introduction section.

4. The study is based on data from stroke centers in the Slovak Republic, which may not be representative of other regions with different climates or healthcare systems.

Thank you for this important comment. We acknowledge the regional specificity of our data, but we would like to emphasize that the stroke centers in our study cover all four climate types present in Slovakia, as classified by the Köppen-Geiger system. This diversity within a single country provides a broad range of meteorological conditions, which could offer insights applicable to other regions with similar climates.

5. The five-year period of data collection might not capture long-term trends or rare events that could affect stroke incidence.

Thank you for your comment. We recognize that a longer dataset could potentially capture more rare events or long-term trends. However, given the constraints of available data, our study focuses on exploring patterns and associations within the five-year timeframe.

6. Findings may not be applicable to populations outside the Slovak Republic, particularly those in regions with significantly different meteorological conditions.

Thank you for highlighting this limitation. We have added a statement in the revised manuscript to acknowledge that our findings may be most relevant to regions with similar climatic conditions to Slovakia and may not be directly applicable to regions with significantly different meteorological conditions.

7. The quality and consistency of the meteorological and stroke data across different centers might vary, potentially introducing bias.

Regarding the meteorological data, we utilized official datasets from national meteorological services, which are consistent and standardized across the country. However, we acknowledge that there could be variations in stroke data collection practices across different centers. However, given that the selected stroke centers are the major one, we are rather confident that the data collection practices are homogeneous across the centers.

8. The study may not adequately account for other factors influencing stroke incidence, such as air pollution, socioeconomic status, or healthcare access.

Thank you for this insight. We have addressed the issue of air pollution in a previous revision. As for healthcare access, the selected stroke centers are major hospitals located in urban areas, minimizing the potential bias related to healthcare accessibility. However, we acknowledge that other factors such as socioeconomic status were not included in our study, which we now explicitly state as a limitation.

9. By concentrating on extreme stroke events (90th percentile), the study may overlook insights into more common stroke occurrences, which could provide a broader understanding.

Thank you for this observation. Initially, our study focused on forecasting raw stroke counts, but the results indicated that meteorological data did not significantly improve forecasting accuracy. To enhance the clinical relevance of our predictions, we shifted our focus to extreme stroke events, which are critical for resource allocation and management. While this approach may overlook more common occurrences, it allows us to address specific, high-impact scenarios in healthcare.

10. The transformation of stroke counts into a binary outcome (high vs. ordinary) might oversimplify the complex nature of stroke incidence, potentially missing subtler trends.

We understand the concern about potential oversimplification. However, as explained in response to point 9, our binary approach was motivated by the need for practical and interpretable predictions in a clinical setting. This approach facilitates the use of sensitivity and specificity metrics, which are more intuitive for clinicians compared to other forecasting metrics such as the Symmetric Mean Squared Error that are used to quantify forecasting accurracy in count time series.

11. The limited predictive accuracy of all three methods suggests that the models may not be well-suited for forecasting extreme stroke events, indicating a need for alternative approaches.

We agree that exploring alternative forecasting methods could be valuable. Our study used Generalized Linear Regression, Random Forest for Time Series, and Croston's method as a starting point. We have made our data publicly available, inviting other researchers to test different approaches that may improve predictive accuracy.

12. Croston’s method, which relies solely on historical stroke data, may not capture the influence of changing meteorological conditions over time.

Indeed, Croston's method was used as a baseline reference due to its univariate nature. While it does not incorporate external factors like meteorological data, it provides a useful benchmark against which to measure the performance of more complex models that do incorporate such data.

13. The use of multivariate logistic regression and Random Forest models could lead to overfitting, particularly if the models are overly complex relative to the size of the dataset.

We acknowledge the risk of overfitting. To mitigate this, we used validation techniques to ensure that the model's performance was evaluated on unseen data.

14. The study may not have identified the most relevant meteorological variables influencing stroke incidence, leading to weak correlations.

In selecting relevant meteorological variables, we used the Akaike Information Criterion (AIC) for the GLM and the Variable Importance algorithm for Random Forest. These methods are well-established for feature selection, and we are confident that they have identified the most relevant variables influencing stroke incidence within the dataset available to us.

15. The daily resolution of meteorological data might be insufficient to capture finer-scale weather patterns that could impact stroke risk.

Thank you for raising this point. We acknowledge that finer temporal resolution could reveal additional insights. However, stroke data at finer resolutions (e.g., hourly) are not available in Slovakia. Our study, therefore, used the best available data to explore the relationship between daily meteorological conditions and stroke incidence.

16. The study does not appear to consider potential temporal lags between changes in weather conditions and stroke incidence, which could affect the accuracy of the predictions.

We have accounted for temporal lags in our analysis. Formula (1) in the manuscript outlines the inclusion of lagged weather time series in our models, both for the Generalized Linear Model and Random Forest for Time Series. This ensures that delayed effects of weather on stroke incidence are considered in our predictions.

17. The complexity of Random Forest and logistic regression models may make it difficult to interpret the relationship between specific meteorological factors and stroke incidence.

We agree that the interpretability of Random Forest models can be challenging. However, the logistic regression model offers interpretable odds ratios, which we have used to discuss the impact of specific meteorological factors on stroke incidence. We acknowledge the trade-off between model complexity and interpretability and have aimed to balance these considerations in our analysis.

18. The models were developed and tested on the same dataset, which may limit the validity of the findings when applied to other datasets or populations.

This is true for our modeling study. However, in the sequential forecasting study, we employed a walk-forward validation approach, where models were trained on data up to time

t and then used to forecast for time t+1. This methodology helps to mitigate the issue of overfitting to a specific dataset, providing a more robust evaluation of the model's predictive capabilities.

19. While the study discusses seasonality, the forecasting models do not explicitly incorporate seasonal patterns, which could be a significant factor in stroke incidence.

The absence of explicit seasonality incorporation in the models is due to the lack of clear seasonal patterns in the data. We conducted an exploratory analysis to detect seasonality, but it was not sufficiently pronounced to justify its inclusion in the models.

20. The study focuses on meteorological factors and may overlook other critical factors, such as lifestyle changes, that could influence stroke incidence.

We agree that factors such as lifestyle changes could influence stroke incidence. However, our study focuses on the relationship between weather conditions and stroke incidence within a specific five-year period. During this time, no significant lifestyle changes were reported in Slovakia, which is why they were not included in our models.

21. The number of extreme stroke events (surpassing the 90th percentile) may be small, leading to challenges in building robust predictive models.

We acknowledge that the small number of extreme events poses a challenge for model robustness. This limitation is inherent to studying rare events, but we believe that focusing on these extreme cases provides valuable insights, particularly for healthcare planning and resource allocation.

22. The binary prediction approach may result in a high rate of false positives or false negatives, reducing the practical utility of the forecasts.

Thank you for this observation. The binary classification approach was chosen to simplify the interpretation of results and make the forecasts more actionable in a clinical setting. We agree that a high rate of false positives or false negatives could reduce the utility of the forecasts. Our analysis shows that meteorological data alone do not provide strong predictive information for extreme stroke events, which contributes to the observed rates of misclassification. This highlights the need for further research to identify additional factors that may improve prediction accuracy. 

23. Using predictive models to forecast extreme medical events could raise ethical concerns, particularly if the forecasts are not accurate enough to inform effective interventions.

Thank you for bringing up this concern. The primary objective of our study is to develop predictive models that can forecast extremely high stroke counts, thereby aiding clinic administrators in better allocating human resources, such as clinicians, to meet potential surges in demand. The use of these forecasts is intended for logistical and planning purposes, not for making direct clinical decisions about individual patient care. Therefore, we believe that the ethical risks are minimal in this context. Our forecasts are designed to inform staffing and resource management, ensuring that clinics are better prepared for potential increases in stroke cases, rather than guiding treatment decisions. We have clarified this objective and its implications in the revised manuscript.

---

Thank you once again for your insightful comments and suggestions regarding the limitations of our study. We greatly appreciate the time and effort you put into reviewing our manuscript. In response to your feedback, we have carefully revised the Discussion section to incorporate most of the limitations you pointed out. These additions have helped us provide a more balanced and comprehensive evaluation of our findings, ensuring that the manuscript more accurately reflects the scope and implications of our study.

---

## [Decision Letter · Decision Letter 3]

23 Aug 2024

Forecasting extremely high ischemic stroke incidence using meteorological time serie

PONE-D-23-38480R3

Dear Dr. Nosal,

We’re pleased to inform you that your manuscript has been judged scientifically suitable for publication and will be formally accepted for publication once it meets all outstanding technical requirements.

Kind regards,

Jyotir Moy Chatterjee

Academic Editor

PLOS ONE

Additional Editor Comments (optional):

Reviewers' comments:

Reviewer's Responses to Questions

**Comments to the Author**

1. If the authors have adequately addressed your comments raised in a previous round of review and you feel that this manuscript is now acceptable for publication, you may indicate that here to bypass the “Comments to the Author” section, enter your conflict of interest statement in the “Confidential to Editor” section, and submit your "Accept" recommendation.

Reviewer #4: All comments have been addressed

Reviewer #5: All comments have been addressed

2. Is the manuscript technically sound, and do the data support the conclusions?

Reviewer #4: Yes

Reviewer #5: Yes

3. Has the statistical analysis been performed appropriately and rigorously? 

Reviewer #4: Yes

Reviewer #5: Yes

4. Have the authors made all data underlying the findings in their manuscript fully available?

Reviewer #4: Yes

Reviewer #5: Yes

5. Is the manuscript presented in an intelligible fashion and written in standard English?

Reviewer #4: Yes

Reviewer #5: Yes

6. Review Comments to the Author

Reviewer #4: I am satisfied with the corrections applied by the authors. I think it has eventually become a very good manuscript, ready for publication.

Reviewer #5: The authors have successfully addressed the majority of the comments. The manuscript is now suitable for acceptance and publication.

7. PLOS authors have the option to publish the peer review history of their article (what does this mean?). If published, this will include your full peer review and any attached files.

Reviewer #4: No

Reviewer #5: No

---

## [Editor Report · Acceptance letter]

30 Aug 2024

PONE-D-23-38480R3 

PLOS ONE

Dear Dr. Nosal, 

I'm pleased to inform you that your manuscript has been deemed suitable for publication in PLOS ONE. Congratulations! Your manuscript is now being handed over to our production team.

Kind regards, 

on behalf of

Mr. Jyotir Moy Chatterjee 

Academic Editor

PLOS ONE